# The Source of Fracture-Cave Mud Fillings of the Ordovician Yingshan Formation and Its Paleokarst Environment in the Northern Slope of the Tazhong Uplift, Tarim Basin, China: Based on Petrology and Geochemical Analysis

Yong Dan [1,2], Guoquan Nie [1,2,*], Bin Liang [1,2], Qingyu Zhang [1,2], Jingrui Li [1,2], Hongqi Dong [1,2] and Shaocong Ji [1,2]

1 Institute of Karst Geology, Chinese Academy of Geological Sciences, Guilin 541004, China; danyong@mail.cgs.gov.cn (Y.D.); liangbin@mail.cgs.gov.cn (B.L.); zqingyu@mail.cgs.gov.cn (Q.Z.); lijingrui@mail.cgs.gov.cn (J.L.); donghongqi@mail.cgs.gov.cn (H.D.); jishaocong@mail.cgs.gov.cn (S.J.)
2 Key Laboratory of Karst Dynamics, MLR, Guilin 541004, China
* Correspondence: nieguoquan@mail.cgs.gov.cn; Tel.: +86-137-3773-4639

**Abstract:** The karst fracture-cave oil and gas reservoirs of the Yingshan Formation in the northern slope of the Tazhong Uplift are well developed and have achieved good exploration results. However, the karst fracture-cave near the top of the Yingshan Formation is basically filled with mud fillings, which seriously affect the reservoir property, and the source and filling environment of the mud fillings have been unclear. Through the petrological and geochemical analysis of the fracture-cave fillings system in the typical wells of the Yingshan Formation, it has been found that (1) the fracture-cave fillings are mainly composed of a mixture of the bedrock dissolution dissociation particles, clay minerals, and calcite cements of the Yingshan Formation, and the content of each component in the different wells or in the cave interval is quite different. (2) Rare earth element analysis shows that the rare earth distribution pattern of the fracture-cave fillings is similar to the bottom marlstone of the Lianglitage Formation, indicating that the fracture-cave fillings should be mainly derived from the early seawater of the deposition during the Lianglitage Formation. (3) Cathodoluminescence, trace element analysis, and previous studies have shown that the formation and fillings of the fractures and caves mainly occurred in the hypergene period, which had the characteristics of an oxidized environment, and that there are two filling effects. First, the limestone of the Yingshan Formation experienced the formation of karst caves due to meteoric freshwater dissolution during the exposure period, and the limestone of the Yingshan Formation was dissolved, resulting in some insoluble clay and residual limestone gravel particles brought into the cave by the meteoric freshwater for filling. Second, the seawater transgression also played an important role during the deposition of the Lianglitage Formation. The clay content in the seawater was high during the early deposition of the Lianglitage Formation, which led to the clay being brought into the caves by the seawater during the deposition of the Lianglitage Formation for further filling; at the same time, calcite deposited into the caves with the clay. The above research promotes the study of the formation mechanism of the karst cave reservoir in the Yingshan Formation and has important theoretical significance for the guiding of the next oil and gas exploration in this area.

**Keywords:** Yingshan Formation; paleokarst; fracture-cave reservoir; geochemistry rare earth element; cathodoluminescence

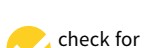

## 1. Introduction

Carbonate karstification can form rich pores, caves, and fractures [1], in which the buried subsurface can become a high-quality reservoir space for oil and gas [2]. In recent decades, karst reservoirs have attracted continuous attention from scientists worldwide [3–8]. Karst reservoirs are also the most important oil and gas exploration fields in

the Tarim Basin. In the early stage, fruitful results were obtained in the exploration of the Lower-Middle Ordovician in the Tabei Uplift and the Tazhong Uplift [9–13]. In recent years, some studies have found transient, exposed karst reservoirs in the exploration of the Yingshan Formation of the Lower-Middle Ordovician in the northern slope belt, based on the discovery of the typical, buried hill karst reservoirs in Tazhong Uplift [14–18]. Studies have shown that the formation of the karst reservoirs in the Yingshan Formation in this area is mainly related to the short-term exposure of the karst in the Caledonian period, and a large number of fracture-cave reservoirs were formed by karstification [19–21]. However, the exploration practice shows that the fracture-cave fillings are widespread in this area. The karst fracture-cave near the top of the Yingshan Formation is basically filled by mud fillings, while the better reservoirs are distributed below it, and the filling effect affects the reservoir property [7,22]. The source, the filling period, and the formation environment and its relationship with the karstification of these mud fillings have always been the difficulties in the reservoir research in this region.

Petrological and geochemical studies on fracture-cave fillings are helpful in understanding the filling process, filling fluid properties, and the filling paleoenvironment [23–26]. Previous studies have explored the genesis of the fracture-cave fillings in the Tazhong Uplift. Wu et al. [27], for example, carried out a petrological and geochemical analysis of the cave fillings of the Yingshan Formation in the YB7 well, indicating that the Ordovician cave fillings in the Tazhong Uplift were derived from the Devonian system [27]. Zhang et al. [22] showed by carbon and oxygen isotope methods that the mud fillings in the karst cave were not from the Yingshan Formation in the northern slope of the Tazhong. Dan et al. [20,28] analyzed the cave fillings in the Yingshan Formation by electron probe, cathodoluminescence, and carbon and oxygen isotope methods [20,28]. It is considered that these fillings were formed in a mixed-water environment during the eogenetic karst period. Although detailed research has been carried out in the region, more samples and analytical methods are needed to analyze the specific sources and the formation environment of the mud fillings in the region during the short exposure period.

Therefore, this paper further analyzed and compared the petrological identification, the main trace elements and the rare earth elements from many wells of the fracture-cave mud fillings of the Yingshan Formation. This study provides new evidence for further understanding of the filling effect in the short exposure period and reservoir formation and evolution, which is of great significance to the guiding of the exploration and development of karst reservoirs in this area.

## 2. Geological Setting

### 2.1. Regional Geological Background

The Tazhong Uplift is located in the central Tarim Basin in northwestern China (Figure 1a). Its western part is connected with the Bachu Fault Uplift, and its eastern part is connected with the Tadong Uplift. The north is close to the Manjiaer depression, and the south is the Tangguzibasi depression, which is northwest-trending [9,29]. The Tazhong Uplift from north to south can be divided into three sub-structural zones, namely the northern slope, the central uplift, and the southern slope [19]. The first exploration in the Tazhong area was of the Middle Ordovician karst reservoir in the central uplift, which is directly covered by the Carboniferous (Figure 1b). The exposure time is greater than 100 My, and the karst time is long, forming a karst buried hill reservoir (Figure 1). In recent years, the exploration has been concentrated in the northern slope zone, which is also the research area of this paper. A set of calcarenite and micrite of carbonate open platform facies was deposited in the Middle-Lower Ordovician Yingshan Formation in this area [30]. At the end of Middle Ordovician, affected by the tectonic movement in the Middle Caledonian period, Episode I, the carbonate rocks of the Yijianfang Formation and the Yingshan Formation, which were newly deposited, were uplifted as a whole, resulting in the exposure of the carbonate platform and strong karstification [31,32] (Figure 2). After the exposure of the carbonate rocks, it was not until the Late Ordovician, during the deposition of the

Lianglitage Formation, that the seawater slowly invaded from east to west again. A set of grayish-green marlstone of mixed continental shelf facies was deposited at the bottom of the Lianglitage Formation, which was a result of transgressive deposition [33].The strata in the high part of the uplift are deeply eroded, and most of them lack the Middle Ordovician Yijianfang Formation and the Tumuxiuke Formation at the bottom of the Upper Ordovician, while the Yingshan Formation of the Lower Ordovician is also eroded, and the amount of erosion increases gradually from north to south. The exposure duration of the Yingshan Formation is about 7–8 My [20], forming the current unconformity (Figures 1b and 2). The Yingshan Formation can be divided into four members, of which the first and second members are relatively pure limestone with a thickness of about 300 m. The third and fourth members are dolomitic limestone and dolomite (Figure 2). The exposed strata are limestones of the first and second members, developing a large number of fractures and caves during the exposure weathering and transgression during the deposition of the Lianglitage Formation, which is an important oil and gas reservoir in the Tarim Basin [7].

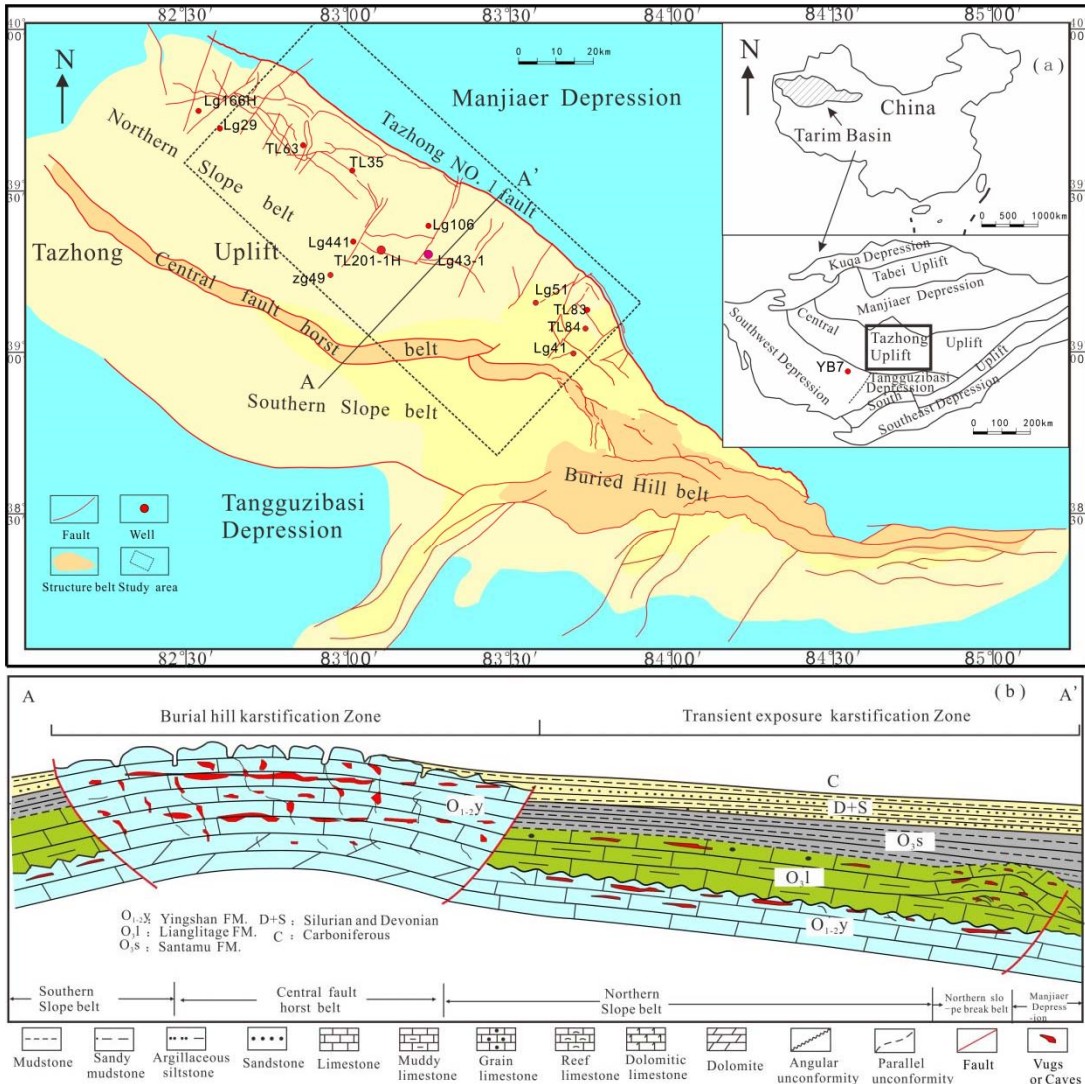

**Figure 1.** Location and geology of study area. (**a**) Tectonic division of Tazhong Uplift and location of study area [19]; (**b**) karstification division of Tazhong Uplift and location of northern slope belt.

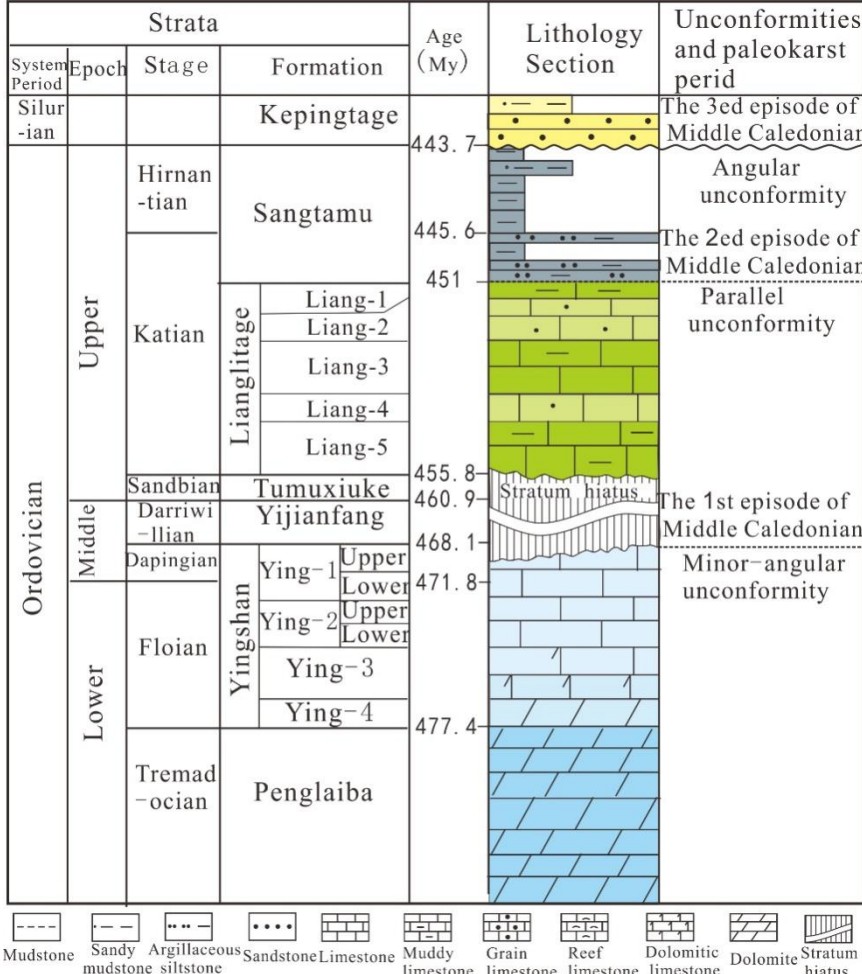

**Figure 2.** Generalized stratigraphy and associated reservoir intervals (northern slope belt) of the Yingshan Formation in the study area. The stages and ages are from the International Commission on Stratigraphy (2014).

### 2.2. Development Characteristics of Karst Cave Mud Fillings in the Northern Slope of the Tazhong Uplift

Dan et al. [20] identified mud-filling and unfilled intervals of 90 wells in the area, based on core, imaging logging, conventional logging interpretation, and drilling and logging methods and found that there are 61 mud-filling intervals [20]. These mud-filling intervals often show high GR (Gamma-ray log) intervals (the GR value is generally 75–150 API) on logging curves (Figure 3). It was found that 58.07% of all the 61 mud-filling intervals developed at 0~40 m below the unconformity (Figure 4). The number of filled intervals decreased with the increase in depth, reaching a low value of 9.68% at 40~60 m, and increased to a peak of 16.13% at 80~100 m after 60 m, while the number of filled intervals decreased sharply over 100 m. According to the statistics of the unfilled caves, the unfilled caves are not developed in 0~20 m under unconformity, relatively developed after 20 m, and less developed over 200 m (Figure 4). The above shows that the caves near the unconformity surface are mostly filled by mud fillings, which also reflects that the filling has a great influence on the reservoir exploration of the Yingshan Formation in the northern slope of Tazhong, and the filling is stronger near the unconformity surface than the lower part. Therefore, studying the source and filling environment of the mud fillings are of great significance for understanding the filling process of oil and gas reservoirs and predicting the distribution of unfilled reservoirs in this area.

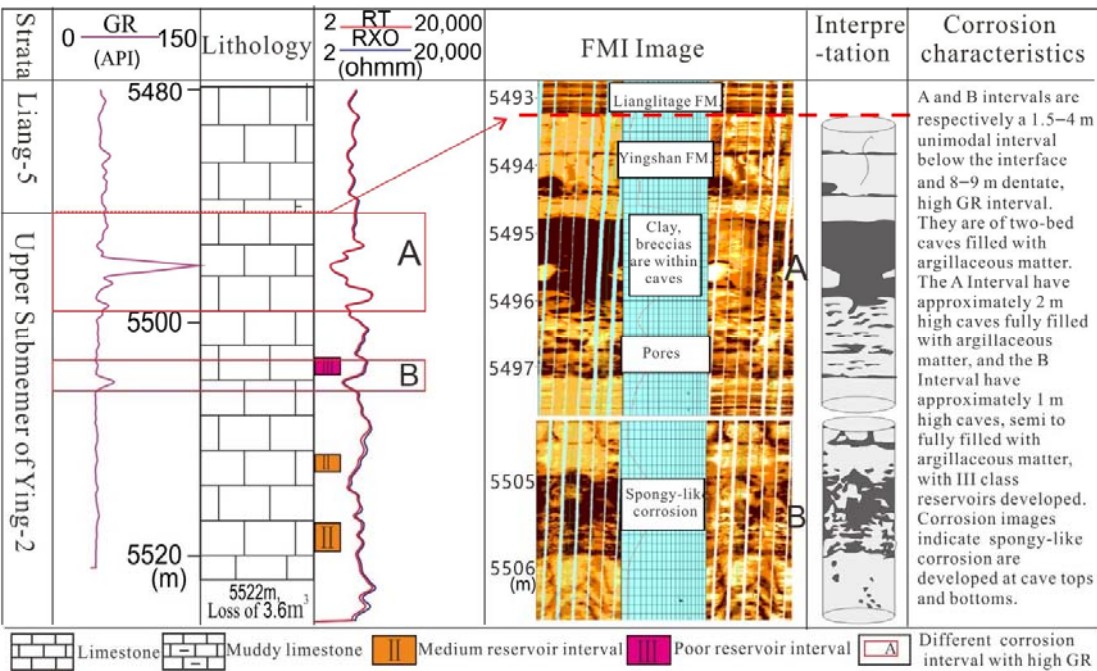

**Figure 3.** The unimodal (or unidentate) high-GR intervals indicate a spongy dissolution zone, which is composed of one karst cave and solution pores in Well LG441. RXO: flushed zone formation resistivity log; GR: gamma-ray log and RT: resistivity log.

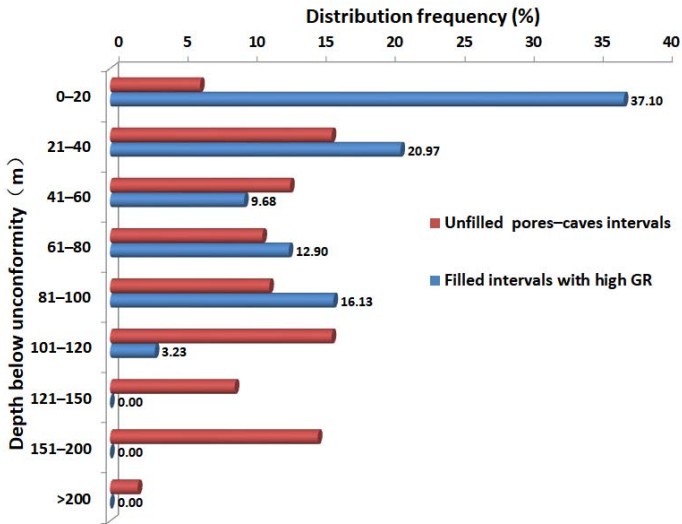

**Figure 4.** Distribution characteristics of mud-filled caves and unfilled caves.

## 3. Methods

### 3.1. Sample Collection

Among the 61 mud-filling intervals identified in this area, only the 3 mud filling intervals of TZ201-1H and ZG43–1 obtains the cores. Therefore, the sample of mud fillings were taken from two wells, TZ201-1H and ZG43–1 (Figure 1a), in the northern slope of the Tazhong Uplift. A cave was developed in the core section of LG43-1, from 5288.5–5289.1 m, and the cave height was 0.6 m (Figure 5). Seven samples were taken continuously from the cave and its surrounding rock and nearby fissure in Well LG43-1. Two caves were developed in the core section of TL201-1H, with a Cave I depth of 5445.0–5445.4 m and a cave height of more than 0.4 m (no cave top in the core section) and a Cave II depth of 5460.5–5460.9 m and a cave height of 0.4 m (Figure 6). Ten samples were taken continuously from the cave and nearby fissure and its surrounding rock in Well TL201-1H. In addition,

marlstone samples were taken from the bottom of the Lianglitage Formation in 11 wells (Figure 1a) in the northern slope of Tazhong, and one sample with high clay content was taken from each well for comparative analysis, in order to study the source of the mud fillings in the karst caves.

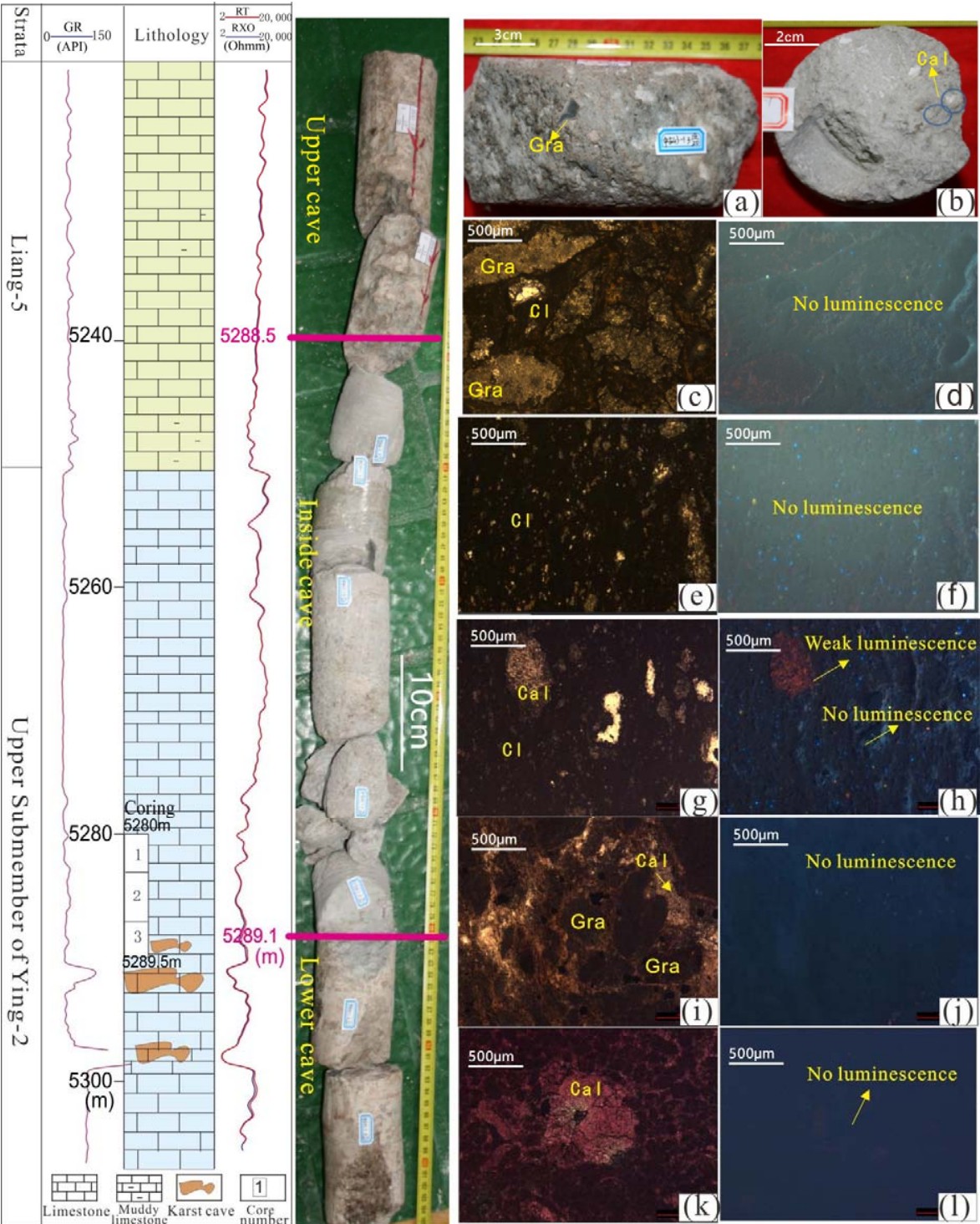

**Figure 5.** The petrological characteristics of karst pore and cave fillings of the Yingshan Formation in Well LG43-1. (**a**,**b**) Karst cave fillings show limestone gravel (Gra), clay (Cl), and calcite (Cal), and the fillings are relatively soft; (**c**,**e**,**g**,**i**) limestone debris grains (Gra), clay (Cl), calcite (Cal), and karst caves. Generally, fillings in karst caves do not show luminescence (**d**,**f**,**h**,**j**) and minor karst caves do not show luminescence (**h**), 5288.5–5289.1 m; (**k**,**l**) calcite in pores in the lower part of karst caves did not show luminescence, 5289.2 m.

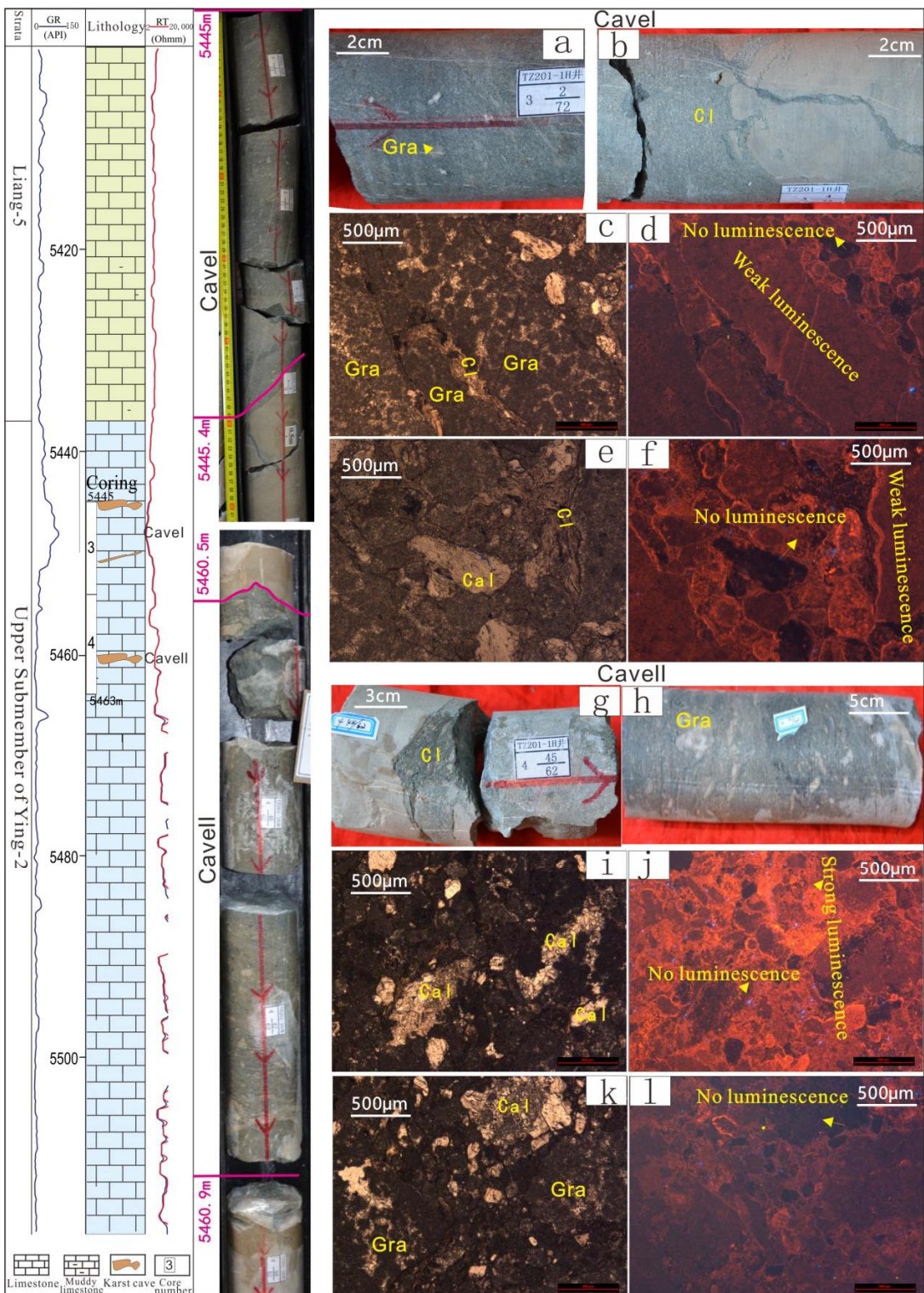

**Figure 6.** The petrological characteristics of karst pore and cave fillings of the Yingshan Formation in Well TL201-1H. (**a**) Cave I fillings, gravel, clay, fillings cemented compactly; (**b**) boundary between bottom of cave I and limestone; (**c,e**) under the microscope of fillings in cave I, bright crystal sand debris limestone particles, calcite and a small amount of clay can be seen; (**d,f**) under cathodoluminescence, the bright crystal sand-chip limestone particles in the filling weakly glow, and calcite does not glow; (**g,h**) fillings at the top and inside of cave II can be seen in limestone gravel, and the fillings are cemented compactly; (**i,k**) the fillings in cave II (limestone particles, calcite, and a small amount of clay); (**j,l**) under cathodoluminescence, the limestone particles in the fillings are weak–strong luminescence, and the luminescence ring is seen at the edge of the particles, while the calcite does not luminescence.



Thin-section identification, sporopollen identification, the main and trace elements test, the rare earth element test, and the clay minerals test were carried out on the samples of the fracture-cave fillings and nearby bedrock in the Yingshan Formation. On the basis of the thin-section identification, samples were selected for cathodoluminescence analysis. The rare earth element test of marlstone at the bottom of the Lianglitage Formation and the testing and analysis methods are described in the following:

### 3.2. Rock Thin-Section Preparation and Identification

The rock sample was bonded to a glass slide with a non-luminous epoxy resin and ground to a thickness of 0.03–0.06 mm. To meet the requirements of the cathodoluminescence and electron probe analysis, the surface of the standard thin section was then highly polished. The thin sections were analyzed under optical plane-polarized light at magnifications ranging from 50× to 500×. A petrographic examination and a sporopollen identification were performed using a Leica Imager microscope-photometer.

### 3.3. Cathodoluminescence Analysis

Following identification from the thin section, the cathodoluminescence analysis focused on the eight fillings in the caverns. The same test conditions, using a beam voltage of 17 KV and a beam current of 500 μA, were performed on all samples. The instrument used for the cathodoluminescence analysis was a CL8200 MK5 cathodoluminescence analyzer produced by CITL in the United Kingdom (in conjunction with a Leica polarizing microscope).

### 3.4. Major and Trace and Rare Earth Elements Analysis

All the samples were crushed and powdered down to a grain size smaller than 200-mesh for geochemical analyses. A total of 500 mg powder was obtained for each sample. Approximately 200 mg of the powdered samples was analyzed for major and trace and rare earth elements. The major elements were measured using an X-ray fluorescence spectrometer (AB-104L), with an analytical precision better than 2%. For the trace elements and rare earth elements, the samples were cleaned in ultra-pure water before dissolution in 2 mL of 15 N double-distilled $HNO_3$. The solutions were then spiked with 10 ppb of internal standards for NexION300D ICP-MS (ELAN DRC-e) analyses with an analytical precision better than 5%. The experimental procedures have been described in other publications [34].

### 3.5. X-ray Diffraction Analyses for Clay Mineral Composition

LG43-1 and TL201-h fracture-cave mud fillings were taken, respectively. The clay minerals in the fracture-caves were extracted by the suspension method. Ten grams of the sample was poured into 50 mL of distilled water and stirred evenly. The sample was kept for 6–8 h. The clay was suspended in water and the other substances were settled. The decanted water was extracted and centrifuged to obtain the clay minerals. The clay minerals were uniformly smeared on the glass sheet to prepare the experimental sample. The X-ray diffraction test of the clay mineral components was divided into three steps: (1) The sample was placed in air for about 12 h and then the natural air-dried directional sample (sheet N) was obtained and analyzed by a diffractometer. (2) The sample was placed in an oven containing ethylene glycol for more than 7 h (40–50 °C) to obtain an ethylene glycol saturated sheet (sheet EG), which was then analyzed by a diffractometer. (3) The sample was heated to 550 °C in a muffle furnace and kept at constant temperature for 2 h to obtain a heating plate (sheet T), which was put into a diffractometer for analysis. It was mainly obtained by D8Advance X-ray diffractometer.

The analyses, as outlined above, were conducted at the State Key Laboratory of Oil and Gas Reservoir Geology and Exploitation, Chengdu University of Technology, Chengdu, China.

## 4. Results

### 4.1. Petrology and Cathodoluminescence Characteristics of Mud Fillings

The LG43-1 karst cave filling interval: the karst cave filling interval is 38 m below the unconformity surface. Core observation and thin-section identification show that it is mainly filled with lime-green mud and breccia and bubbling with 5% dilute hydrochloric acid (Figure 5). The filling is mainly composed of gravel particles, clay, and a small amount of calcite cement formed by limestone debris dissolution. Among them, the breccia components were mainly micrite and lithic limestone, by microscopic identification (Figure 5c,i). The clay composition was analyzed by X-ray diffraction, and the clay minerals were mainly an illite and illite–smectite mixed layer (Table 1). In addition, calcite settles between these substances (Figure 5b). Quartz, feldspar, and other terrestrial minerals were not found, and pollen was not found. On the whole, the content of the detrital particles or gravels composed of limestone debris gravel is about 40–50%; the content of clay minerals is about 32–38%; and the content of calcite cement is 5–8%. The limestone detrital particles or gravels are of different sizes and poorly sorted, and the degree of roundness is subangular–subcircular (Figure 5a). Cathodoluminescence shows that the limestone particles and clay of the fillings in the cave do not emit light as a whole, and only a small amount of the calcite has weak luminescence (Figure 5d,f,h,j,l).

**Table 1.** Analysis results of clay mineral composition of karst cave mud fillings.

| Well Number | Depth | Lithology | Relative Content of Clay Minerals (%) | | | | | |
|---|---|---|---|---|---|---|---|---|
| | | | K | C | I | S | I/S | C/S |
| LG43-1 | 5288.6 m | Karst cave mud fillings | / | / | 10 | / | 90 | / |
| TL201-1h | 5445.1 m | Karst cave mud fillings | 1 | 3 | 9 | / | 75 | 12 |

Note: K: Kaolinite, C: Chlorite, I: Illite, S: Smectite, I/S: Illite–Smectite mixed layer, C/S: Chlorite–Smectite mixed layer.

TL201-1H cave filling interval: two cave filling intervals were found in the coring of the well, located at 8 m and 23 m below the unconformity. The core observation shows that the lithology of the two karst cave fillings is similar, but different from that of the LG43-1 karst cave section. The mud fillings of LG43-1 have low calcium content and are loose, while the mud fillings of TL201-1H have high calcium content and dense cementation (Figure 6a,b,g,h). Microscopic identification of the TL201-1H fillings shows that they are mainly composed of mud microcrystalline or sandy limestone debris composed of debris particles or gravel (about 70–80%), clay minerals (about 10–15%) and calcite cement (about 10–15%) (Figure 5c,e,i,k); the filling-clay content is lower than that of LG43-1; the clay composition by X-ray diffraction analysis is mainly an illite–montmorillonite mixed layer of chlorite, illite, and some kaolinite, which is slightly different from that of Lg43-1 (Table 1). The cathodoluminescence shows that the limestone particles (regardless of the micrite particles and the sparry limestone particles) in the cave have medium-strong luminescence, and have a luminescence ring, while the clay and calcite do not emit light (Figure 6d,f,j,l).

### 4.2. Main and Trace Element Characteristics

#### 4.2.1. Main Elements

The CaO content in the limestone of the Yingshan Formation near the karst caves of the two wells is 54–57%, and the $SiO_2$, $Al_2O_3$, and $Fe_2O_3$ contents are extremely low, with the total content not exceeding 5%. The content of CaO is 30–40%, followed by $SiO_2$, $Al_2O_3$, and $Fe_2O_3$. The main component of the clay minerals is aluminum silicate; so, the content of the clay minerals is approximately equal to the $SiO_2 + Al_2O_3 + Fe_2O_3$. The content of the clay minerals in the cave filling is about 17–39% (Figure 7), which is quite different from the bedrock test results. The CaO content of the TL201-1H cave fillings is high, with an average of 35%, followed by the clay, with an average of 21.65%. The clay content of the LG43-1 karst cave fillings is high, with an average of 37.34%, followed by the CaO, with an average

of 22.31%, indicating that the clay content of the LG43-1 karst cave fillings is higher than that of the TL201-1H karst cave fillings, but the carbonate rock debris is less. The above analysis shows that the types of cave fillings are basically the same, but different types of content in different wells have a large gap, which may be related to the distance between the cave and the unconformity surface, the karst landform, and the filling environment.

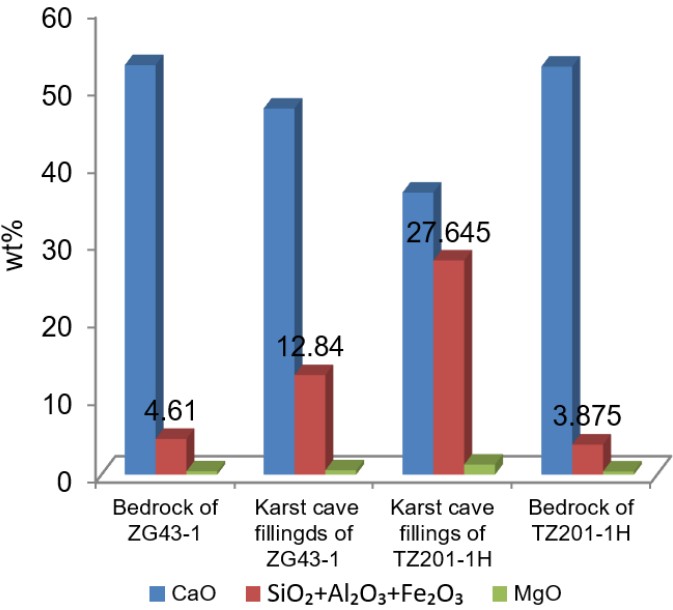

**Figure 7.** Distribution of chemical composition of fillings in the Yingshan Formation caves.

4.2.2. Trace Elements

The trace elements in the fillings were tested for. The results are shown in Table 2. According to previous studies [26,35,36], some trace element ratios can effectively be used to judge the environment. In this study, the trace element ratios Sr/Ba, V/Cr, Ni/Cr, V/V+Ni, and B/Ga were selected as the basis for judging the filling environment.

(1) Sr/Ba: Studies suggest that an Sr/Ba of less than 1 is usually formed in fresh water, greater than 1 is formed in seawater, and a ratio of 0.6–1.0 is formed in brackish water [37]. The Sr content of the two TL201-1H cave fillings is 100–108 ppm, the Ba is 5.2 ppm–7.4 ppm, and the Sr/Ba is 14.59–19.23, much higher than 1 (Table 2), which is consistent with that of the Yingshan Formation limestone bedrock Sr/Ba of 5.18–25.21. The Sr content of the LG43-1 cave fillings is 128–134 ppm, the Ba is 159 ppm–203 ppm, and the Sr/Ba is 0.63–0.84, both less than 1 and between 0.6 and 1.0.

(2) B/Ga: Wang et al. [38] and Zhang et al. [39] summarized the previous data and proposed that the B/Ga ratio of the terrestrial environment is generally less than 3.0–3.3, while the B/Ga ratio of the marine environment is generally greater than 4.5–5.0, and the transition environment is between them. The B/Ga of a cave filling sample in TL201-1H was 8.45, greater than 5.0, and the cave fillings were related to seawater. The B/Ga ratio of a cave filling in LG43-1 was 3.64, which also indicates that the fillings were formed in brackish water or mixed water.

(3) V/Cr, Ni/Cr, and V/V+Ni: V/Cr, Ni/Cr, and V/V+Ni can be used to judge the redox environment at the filling, and it was found that values less than 2, 5, and 0.75 can be judged as that of an oxidation environment [40]. The statistics in Table 1 show that the V/Cr of the cave fillings in Well TL201-1H is 0.54, the Ni/Cr is 0.77, and the V/V+Ni is 0.41. The V/Cr, Ni/Cr, and V/V+Ni of the cave fillings in the LG43-1 well are 0.6–0.77, 0.12–0.14, and 0.83–0.86, respectively. The V/Cr, Ni/Cr, and V/V+Ni of the cave fillings were less than 2, 5, and 0.75, respectively. The fillings were mainly filled in a shallow-water oxidation environment; there was no sample in a deep-water reduction environment.

### 4.3. Characteristics of Rare Earth Elements in Fillings of Yingshan Formation

In theory, the heterogeneity of the composition characteristics of the rare earth elements can provide a basis for the source analysis of the fracture-cave fillings [41–45]. Therefore, in order to understand the source of the mud fillings in the karst caves of the Yingshan Formation, in addition to collecting the mud fillings in nine fractures and caves near the karst caves in the northern slope of the Tazhong Uplift, the mud fillings in the bottom of the Ordovician Lianglitage Formation and the limestone of the Yingshan Formation in other wells were systematically sampled and tested; a total of 18 samples was collected, and the test results are in Tables 3 and 4. According to the test results, the total amount of rare earth (ΣREE), the ratio of light to heavy rare earth (ΣLREE/ΣHREE), the δEu, and the δCe of each sample were calculated (Tables 3 and 4).

(1) Total rare earth (ΣREE): Total rare earth can reflect the relevant information of a fluid source. Among the tested samples of mud fillings, the total rare earth in LG43-1 was 75.3–210.9 ppm, with an average of 146.2 ppm. The total rare earth in TL201-1H was 102.4–137.5 ppm, with an average of 117.0 ppm. The total rare earth in the limestone of Yingshan Formation near the fracture-cave was 13.05–26.21 ppm, with an average of 17.65 ppm. The total rare earth in the bottom of the Lianglitage Formation was 26.7–164.34 ppm, with an average of 49.16 ppm. Except for TL201-1H, the argillaceous strip of marlstone was 164.34 ppm, and the total rare earth in the bottom of the Lianglitage Formation was less than 100 ppm.

(2) Light and heavy rare earth ratio (ΣLREE/ΣHREE): The ratio of light and heavy rare earth elements is an important parameter of rare earth element geochemistry, which reflects the degree of differentiation of the rare earth elements to a certain extent. The LREE/HREE values of the LG43-1 mud fillings ranged from 10.83 to 16.87, with an average of 14.79. The LREE/HREE values of the TZ201-1H fracture-cave mud fillings ranged from 8.87 to 12.63, with an average of 9.76, slightly lower than that of the LG43-1 fracture-cave fillings (Figure 8). The LREE/HREE values of the Yingshan Formation limestone near the fracture-cave were between 7.38 and 10.85, with an average of 9.20. The LREE/HREE values of the limestone at the bottom of the Lianglitage Formation were between 7.35 and 12.38, with an average of 9.35. On the whole, the differentiation degree of the light and heavy rare earth elements of the fracture-cave mud fillings was greater than that of the limestone of the Lianglitage Formation and the Yingshan Formation.

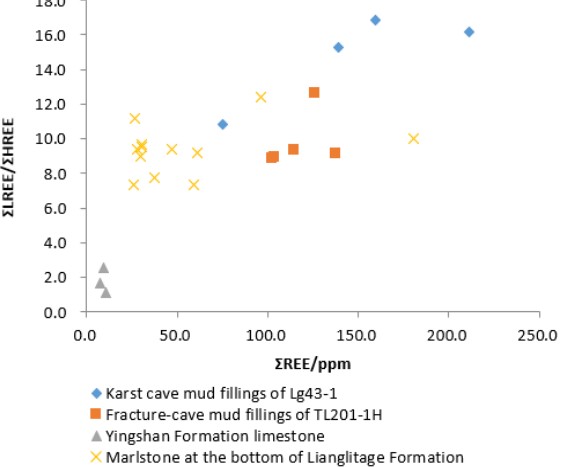

**Figure 8.** Correlation between the total amount of rare earth and the ratio of light to heavy rare earth of the fracture-cave fillings in the northern slope of Tazhong.

**Table 2.** Test results of main and trace elements of fracture-cave fillings and nearby bedrock.

| Sample No. | Depth (m) | Sample Category | SiO2 (wt %) | Al2O3 (wt %) | Fe2O3 (wt %) | CaO (wt %) | MgO (wt %) | K2O (wt %) | Na2O (wt %) | TiO2 (wt %) | P2O5 (wt %) | MnO (wt %) | Ba (ppm) | Cr (ppm) | Ni (ppm) | Sr (ppm) | B (ppm) | V (ppm) | Ga (ppm) | B/Ga | Sr/Ba | V/Cr | Ni/Cr | V/V + Ni |
|---|---|---|---|---|---|---|---|---|---|---|---|---|---|---|---|---|---|---|---|---|---|---|---|---|
| LG43-1 1(11/16) | 5281.3 | Mud filled in the karst fracture | 17.86 | 9.57 | 1.86 | 33.77 | 0.87 | 3.27 | 0.34 | 0.45 | 0.11 | 0.006 | 255 | 58 | 8.45 | 118 | / | / | / | / | 0.46 | / | 0.15 | / |
| LG43-1 2(7/25)9 | 5283.7 | Mud filled in the karst fracture | 45.12 | 25.65 | 2.85 | 3.53 | 1.72 | 8.26 | 0.6 | 1.22 | 0.29 | 0.007 | 416 | 117 | 7.15 | 220 | / | / | / | / | 0.53 | / | 0.06 | / |
| LG43-1 3(11/20) | 5288.4 | The limestone on the top of the cave | 1.67 | 0.91 | 1.06 | 52.02 | 0.24 | 0.32 | 0.12 | 0.045 | 0.013 | 0.002 | 22.2 | 5.89 | 6.03 | 115 | / | / | / | / | 5.18 | / | 1.02 | / |
| LG43-1 3(12/20) | 5288.5 | Karst cave fillings | 21.93 | 12.48 | 1.94 | 29.4 | 0.76 | 4.1 | 0.34 | 0.71 | 0.18 | 0.005 | 203 | 73.7 | 8.94 | 128 | 63 | 44 | 17.3 | 3.64 | 0.63 | 0.6 | 0.12 | 0.83 |
| LG43-1 3(15/20) | 5288.9 | | 25.4 | 14.94 | 1.75 | 25.69 | 0.87 | 4.66 | 0.41 | 0.59 | 0.13 | 0.004 | 195 | 59.3 | 7.44 | 130 | 63 | 44 | 17.3 | 3.64 | 0.67 | 0.74 | 0.13 | 0.86 |
| LG43-1 3(16/20) | 5289 | | 20.14 | 11.71 | 1.72 | 31.3 | 0.72 | 3.75 | 0.34 | 0.55 | 0.13 | 0.005 | 159 | 56.8 | 7.94 | 134 | 63 | 44 | 17.3 | 3.64 | 0.84 | 0.77 | 0.14 | 0.85 |
| LG43-1 3(17/20) | 5289.1 | The limestone at the bottom of the cave | 1.16 | 0.61 | 1.37 | 52.69 | 0.19 | 0.21 | 0.12 | 0.038 | 0.013 | 0.004 | 15.7 | 5.1 | 5.12 | 101 | / | / | / | / | 6.43 | / | 1 | / |
| TL201-1H 3(3/72) | 5445.3 | Karst cave mud fillings | 16.70 | 7.15 | 3.25 | 36.76 | 1.44 | 2.06 | 0.51 | 0.39 | 0.066 | 0.023 | 7.4 | 29.8 | 23 | 108 | 57.2 | 16.5 | 6.77 | 8.45 | 14.59 | 0.55 | 0.77 | 0.42 |
| TL201-1H 3(4/72)a | 5445.4 | | 15.45 | 6.71 | 4.38 | 36.94 | 1.34 | 1.96 | 0.49 | 0.049 | 0.001 | 0.023 | 6.7 | 26.9 | 27.5 | 102 | 57.2 | 16.5 | 6.77 | 8.45 | 15.22 | 0.61 | 1.02 | 0.40 |
| TL201-1H 3(4/72)b | 5445.5 | The limestone at the bottom of the cave | 2.31 | 1.21 | 0.72 | 52.37 | 0.51 | 0.31 | 0.022 | 0.36 | 0.076 | 0.0081 | 4.2 | 2.9 | 5 | 100 | / | / | / | / | 23.81 | / | 1.72 | / |
| TL201-1H 3(26/72) | 5448.3 | The limestone at the bottom of the cave | 1.1 | 0.49 | 0.25 | 55.01 | 0.36 | 0.15 | 0.11 | 0.026 | 0.013 | 0.006 | 11.9 | 4.86 | 5.85 | 124 | / | / | / | / | 10.42 | / | 1.2 | / |
| TL201-1H 3(40/72) | 5450 | Mud filled in the karst fracture between caves | 18.54 | 9.59 | 6.65 | 29.85 | 1.86 | 2.81 | 0.33 | 0.56 | 0.19 | 0.03 | / | / | / | / | / | / | / | / | / | / | / | / |
| TL201-1H 3(47/72) | 5450.9 | The limestone between caves | 5.14 | 2.98 | 1.18 | 48.6 | 1 | 0.89 | 0.16 | 0.16 | 0.021 | 0.011 | / | / | / | / | / | / | / | / | / | / | / | / |
| TL201-1H 3(52/72) | 5451.5 | Mud fillings of fracture between caves | 14.35 | 7.77 | 3.28 | 37.43 | 1.4 | 2.29 | 0.22 | 0.44 | 0.028 | 0.017 | / | / | / | / | / | / | / | / | / | / | / | / |
| TL201-1H 4(44/62) | 5460.5 | The limestone on the top of the cave | 1.53 | 0.74 | 0.18 | 54.26 | 0.21 | 0.24 | 0.11 | 0.027 | 0.009 | 0.002 | / | / | / | / | / | / | / | / | / | / | / | / |
| TL201-1H 4(46/62)a | 5460.6 | Karst cave fillings | 18.09 | 7.80 | 2.86 | 35.87 | 1.26 | 2.27 | 0.72 | 0.42 | 0.058 | 0.018 | 5.2 | 30.8 | 23.6 | 100 | 55.2 | 16.1 | 6.56 | 8.42 | 19.23 | 0.52 | 0.77 | 0.41 |
| TL201-1H 4(46/62)b | 5460.6 | Bedrock of cave wall | 2.08 | 1.19 | 0.24 | 52.96 | 0.29 | 0.24 | 0.018 | 0.047 | 0.001 | 0.0053 | 3.3 | 1.2 | 5 | 83.2 | / | / | / | / | 25.21 | / | 4.17 | / |

Table 3. Results of rare earth elements in mud fillings and nearby limestone of Yingshan Formation in the northern slope of Tazhong (ppm).

| Sample No. | ZG43-1 1(11/16) | ZG43-1 2(7/25) | ZG43-1 2(15/25) | ZG43-1 3(12/20) | TZ201-1H 3(3/72) | TZ201-1H 3(4/72)a | TZ201-1H 3(52/72) | TZ201-1H 4(23/62) | TZ201-1H 4(46/62)a | ZG43-1 1(10/16) | ZG43-1 3(10/20) | TZ201-1H 3(4/72)b |
|---|---|---|---|---|---|---|---|---|---|---|---|---|
| Sample Category | Mud Fillings in Fractures | Fracture-Cave Mud Fillings | Fracture-Cave Mud Fillings | Mud Fillings in Caves | Mud Fillings in Cave I | Mud Fillings in Cave I | Mud Fillings in Fractures | Mud Fillings in Fractures | Mud Fillings in Cave II | The Limestone on the Upper Part of the Cave | The Limestone on the Top of the Cave | The Limestone at the Bottom of the Cave |
| Depth (m) | 5281.3 | 5283.7 | 5284.8 | 5288.5 | 5445.3 | 5445.4 | 5450.5 | 5457.5 | 5460.6 | 5281.2 | 5288.4 | 5445.4 |
| La | 19.5003 | 57.7251 | 48.7969 | 34.5114 | 26.0590 | 22.8466 | 36.2474 | 42.5341 | 23.3390 | 3.2120 | 2.6548 | 5.7555 |
| Ce | 31.9033 | 95.3273 | 66.5996 | 62.6628 | 48.0788 | 41.2997 | 55.2532 | 48.1358 | 41.7673 | 5.4656 | 5.3077 | 11.0119 |
| Pr | 3.5742 | 10.3882 | 7.9902 | 7.1037 | 5.4706 | 5.0679 | 6.4390 | 5.8157 | 5.0443 | 0.6059 | 0.6763 | 1.2333 |
| Nd | 11.6941 | 31.2071 | 24.0243 | 22.8481 | 19.8422 | 18.7547 | 21.6641 | 17.4889 | 19.0321 | 2.1768 | 2.6616 | 4.6639 |
| Sm | 1.9395 | 3.3468 | 2.6423 | 2.8876 | 3.6064 | 3.3757 | 3.6668 | 2.5457 | 3.5167 | 0.3700 | 0.5627 | 0.8227 |
| Eu | 0.3878 | 0.6408 | 0.5139 | 0.4918 | 0.6987 | 0.6827 | 0.6728 | 0.5303 | 0.6754 | 0.1226 | 0.1792 | 0.1926 |
| Gd | 1.7500 | 3.4296 | 2.4241 | 2.5698 | 3.1026 | 2.9052 | 3.2691 | 2.3434 | 2.9316 | 0.3123 | 0.4620 | 0.7618 |
| Tb | 0.2678 | 0.4430 | 0.3545 | 0.3401 | 0.4999 | 0.4723 | 0.5526 | 0.3656 | 0.4795 | 0.0476 | 0.0752 | 0.1199 |
| Dy | 1.5187 | 2.5590 | 2.0493 | 1.8912 | 2.8884 | 2.6858 | 3.3798 | 2.1652 | 2.6862 | 0.2753 | 0.4100 | 0.6544 |
| Ho | 0.3003 | 0.5587 | 0.4269 | 0.3784 | 0.5753 | 0.5321 | 0.7019 | 0.4531 | 0.5299 | 0.0576 | 0.0851 | 0.1283 |
| Er | 0.9121 | 1.8356 | 1.3214 | 1.2116 | 1.6213 | 1.4950 | 2.1417 | 1.4386 | 1.5295 | 0.1592 | 0.2383 | 0.3655 |
| Tm | 0.1847 | 0.3975 | 0.2758 | 0.2412 | 0.3065 | 0.2841 | 0.4203 | 0.2933 | 0.2889 | 0.0306 | 0.0447 | 0.0665 |
| Yb | 1.2523 | 2.6434 | 1.7994 | 1.6668 | 1.8811 | 1.7454 | 2.7101 | 1.9215 | 1.8021 | 0.1897 | 0.2749 | 0.3785 |
| Lu | 0.1863 | 0.3974 | 0.2717 | 0.2551 | 0.2735 | 0.2502 | 0.3975 | 0.2902 | 0.2585 | 0.0290 | 0.0413 | 0.0536 |
| ΣREE | 75.3714 | 210.8996 | 159.4904 | 139.0598 | 114.9042 | 102.3973 | 137.5163 | 126.3213 | 103.8811 | 13.0543 | 13.6737 | 26.2084 |
| ΣLREE/ΣHREE | 10.8281 | 16.1963 | 16.8738 | 15.2562 | 9.3067 | 8.8744 | 9.1316 | 12.6257 | 8.8875 | 10.8523 | 7.3812 | 9.3650 |
| La */Yb * | 1.5085 | 2.1155 | 2.6271 | 2.0058 | 1.3420 | 1.2680 | 1.2957 | 2.1444 | 1.2547 | 1.6404 | 0.9356 | 1.4731 |
| La */Sm * | 1.7910 | 3.0722 | 3.2896 | 2.1289 | 1.2871 | 1.2055 | 1.7608 | 2.9761 | 1.1821 | 1.5465 | 0.8404 | 1.2461 |
| δEu | 0.9241 | 0.8290 | 0.8915 | 0.7927 | 0.9167 | 0.9567 | 0.8531 | 0.9532 | 0.9225 | 1.5828 | 1.5410 | 1.0681 |
| δCe | 0.8232 | 0.8374 | 0.7194 | 0.8681 | 0.8742 | 0.8348 | 0.7772 | 0.6385 | 0.8366 | 0.8457 | 0.8627 | 0.8980 |

Note: * Representing standardized shale data in North America; δEu = 2Eu */(Sm * + Gd *); δCe = 2Ce */(La * + Pr *).

**Table 4.** Results of rare earth elements in marlstone at the bottom of Lianglitage Formation in the northern slope of Tazhong (ppm).

| Sample No. | TZ201-1H 1(15/67) | ZG49 3(31/55) | TZ35 16(16/47) | TZ83 11(37/47) | ZG29 5(44/65) | ZG41 1(66/72) | ZG41 1(71/72) | TZ63 8(27/51) | TZ84 7(28/53) | ZG106 3(15/75) | ZG166H 1(53/65) | ZG51 5(51/51) |
|---|---|---|---|---|---|---|---|---|---|---|---|---|
| Depth (m) | 5055.5 | 5312 | 5772.5 | 5629 | 6063.7 | 5546 | 5546.3 | 6073.7 | 5541 | 6079.5 | 6226.5 | 5054.8 |
| La | 41.0923 | 21.4998 | 16.8955 | 7.3295 | 6.9799 | 6.8179 | 6.3464 | 2.7336 | 10.8868 | 6.6422 | 10.8961 | 4.9676 |
| Ce | 76.3807 | 42.7084 | 24.7200 | 12.7963 | 12.4056 | 12.9164 | 11.7527 | 5.4355 | 19.7859 | 15.3606 | 23.2400 | 10.5088 |
| Pr | 8.9728 | 4.9938 | 2.7034 | 1.3633 | 1.3476 | 1.4101 | 1.2913 | 0.6266 | 2.1530 | 1.9146 | 3.0279 | 1.2383 |
| Nd | 31.4832 | 16.4583 | 8.6651 | 4.8268 | 4.8431 | 5.2156 | 4.7462 | 2.6535 | 7.9201 | 7.6449 | 11.9993 | 4.8388 |
| Sm | 5.3751 | 2.6294 | 1.4979 | 0.8253 | 0.8684 | 0.9495 | 0.8520 | 0.6200 | 1.4220 | 1.4246 | 2.3344 | 0.9003 |
| Eu | 1.0338 | 0.5605 | 0.3835 | 0.2280 | 0.2133 | 0.2379 | 0.2150 | 0.1857 | 0.3083 | 0.3126 | 0.4949 | 0.2418 |
| Gd | 4.6373 | 2.1456 | 1.3725 | 0.7218 | 0.7583 | 0.8095 | 0.7357 | 0.6186 | 1.1851 | 1.2346 | 1.9617 | 0.8312 |
| Tb | 0.7329 | 0.3171 | 0.2406 | 0.1250 | 0.1301 | 0.1331 | 0.1207 | 0.1131 | 0.1947 | 0.2015 | 0.3290 | 0.1360 |
| Dy | 4.0673 | 1.7086 | 1.4324 | 0.7136 | 0.7354 | 0.7554 | 0.6875 | 0.6857 | 1.1336 | 1.1205 | 1.8411 | 0.8094 |
| Ho | 0.8337 | 0.3335 | 0.3213 | 0.1475 | 0.1528 | 0.1475 | 0.1362 | 0.1405 | 0.2366 | 0.2182 | 0.3693 | 0.1615 |
| Er | 2.3979 | 0.9786 | 0.9628 | 0.4224 | 0.4530 | 0.4154 | 0.3857 | 0.3953 | 0.6725 | 0.6122 | 1.0293 | 0.4670 |
| Tm | 0.4623 | 0.1926 | 0.2128 | 0.0876 | 0.0937 | 0.0797 | 0.0753 | 0.0832 | 0.1313 | 0.1156 | 0.1944 | 0.0867 |
| Yb | 2.8383 | 1.2980 | 1.2277 | 0.5296 | 0.5691 | 0.4830 | 0.4690 | 0.4909 | 0.8545 | 0.6929 | 1.1548 | 0.5164 |
| Lu | 0.4089 | 0.2040 | 0.2075 | 0.0791 | 0.0844 | 0.0690 | 0.0677 | 0.0774 | 0.1222 | 0.0985 | 0.1776 | 0.0765 |
| ΣREE | 180.7164 | 96.0282 | 60.8430 | 30.1958 | 29.6345 | 30.4399 | 27.8813 | 26.8175 | 47.0066 | 37.5935 | 59.0498 | 25.7803 |
| ΣLREE/ΣHREE | 10.0337 | 12.3781 | 9.1784 | 9.6827 | 8.9555 | 9.5234 | 9.4123 | 11.1931 | 9.3755 | 7.7550 | 7.3674 | 7.3575 |
| La */Yb * | 1.4025 | 1.6047 | 1.3332 | 1.3408 | 1.1882 | 1.3674 | 1.3110 | 1.6854 | 1.2343 | 0.9287 | 0.9140 | 0.9319 |
| La */Sm * | 1.3617 | 1.4565 | 2.0091 | 1.5820 | 1.4317 | 1.2791 | 1.3268 | 1.6538 | 1.3637 | 0.8305 | 0.8314 | 0.9828 |
| δEu | 0.9087 | 1.0345 | 1.1741 | 1.2967 | 1.1539 | 1.1909 | 1.1921 | 1.0696 | 1.0417 | 1.0347 | 1.0145 | 1.2273 |
| δCe | 0.8647 | 0.8973 | 0.7783 | 0.8729 | 0.8744 | 0.9038 | 0.8900 | 0.8390 | 0.8847 | 0.9354 | 0.8797 | 0.9228 |

Note: * Representing standardized shale data in North America; δEu = 2Eu */(Sm * + Gd *); δCe = 2Ce */(La * + Pr *).

(3) δEu and δCe anomalies: Eu and Ce will show abnormal valence changes in different environments, thus providing important information on the filling environment. The δCe of all the kinds of samples is less than 1, which is a negative anomaly, while the δEu is less than 1, which is 0.79–0.96, and is a negative anomaly in both wells (Figure 9). The limestone of Yingshan Formation and the marlstone of Lianglitage Formation are normal.

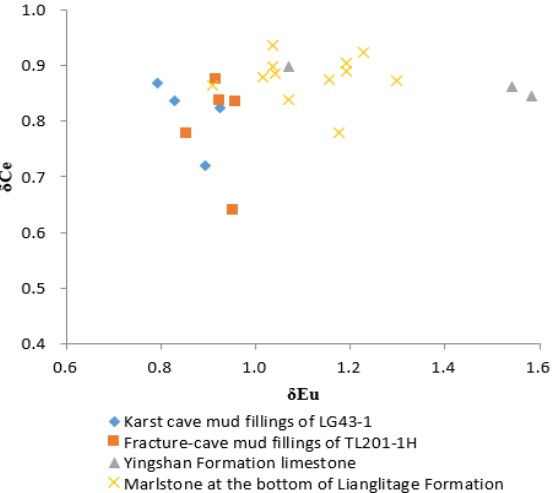

**Figure 9.** Characteristic of δCe and δEu of fracture-cave fillings in the northern slope of Tazhong.

(4) Rare earth distribution pattern: The rare earth element standardization of sedimentary rocks usually uses North American shale (NASC) or post Archean shale (PAAS) as the standard value [46]. In this paper, the NASC standard value, remeasured by Gromet et al. (1984), is used for standardization [47], and then the logarithm of the standardized value is used as the ordinate, and the order of La to Lu is used as the abscissa to make the REE swallow curve (Figure 10). Because the fillings from the different sources have different REE distribution patterns, the nature and source of the filling fluid can be judged by the analysis of the distribution pattern morphology and the comparison of the REE distribution patterns in different strata [48]. The rare earth distribution patterns of different samples are as follows:

**Fillings of Yingshan Formation in the northern slope of Tazhong:** The rare earth distribution pattern of the LG43-1 fracture-cave mud fillings belongs to the right-leaning type. The front of the light rare earth and the back of the heavy rare earth are enriched, especially the La, with an obvious Ce negative anomaly and a slight Eu negative anomaly (δEu average 0.82) (Figure 10a). The distribution pattern of TL201-1h also belongs to the slightly right-leaning type. The front part of the light rare earth and the rear part of the heavy rare earth are enriched, especially the La, with an obvious negative Ce anomaly and a slight negative Eu anomaly (δEu average 0.91) (Figure 10b). On the whole, the rare earth distribution patterns of the mud fillings in the two wells are similar, and the Eu and Ce are negative anomalies, indicating that they are the same source.

**The Yingshan Formation limestone:** The distribution pattern of limestone in the Yingshan Formation belongs to the micro-right-leaning type. The light rare earth is slightly enriched, and the total amount of rare earth is extremely low, an order of magnitude lower than that of the cave filling. It has an obvious positive Eu anomaly (δEu average 1.32), which is different from the mud fillings in the fracture-cave.

**Marlstone at the bottom of the Lianglitage Formation:** It can be divided into two types of distribution patterns. Type I belongs to the right-leaning type. The front part of the light rare earth and the back part of the heavy rare earth are enriched, and there is a slight positive Eu anomaly (δEu average 1.11). Type II belongs to the flat type, and the light and heavy rare earths are equivalent. Ce and Eu have no obvious anomalies.

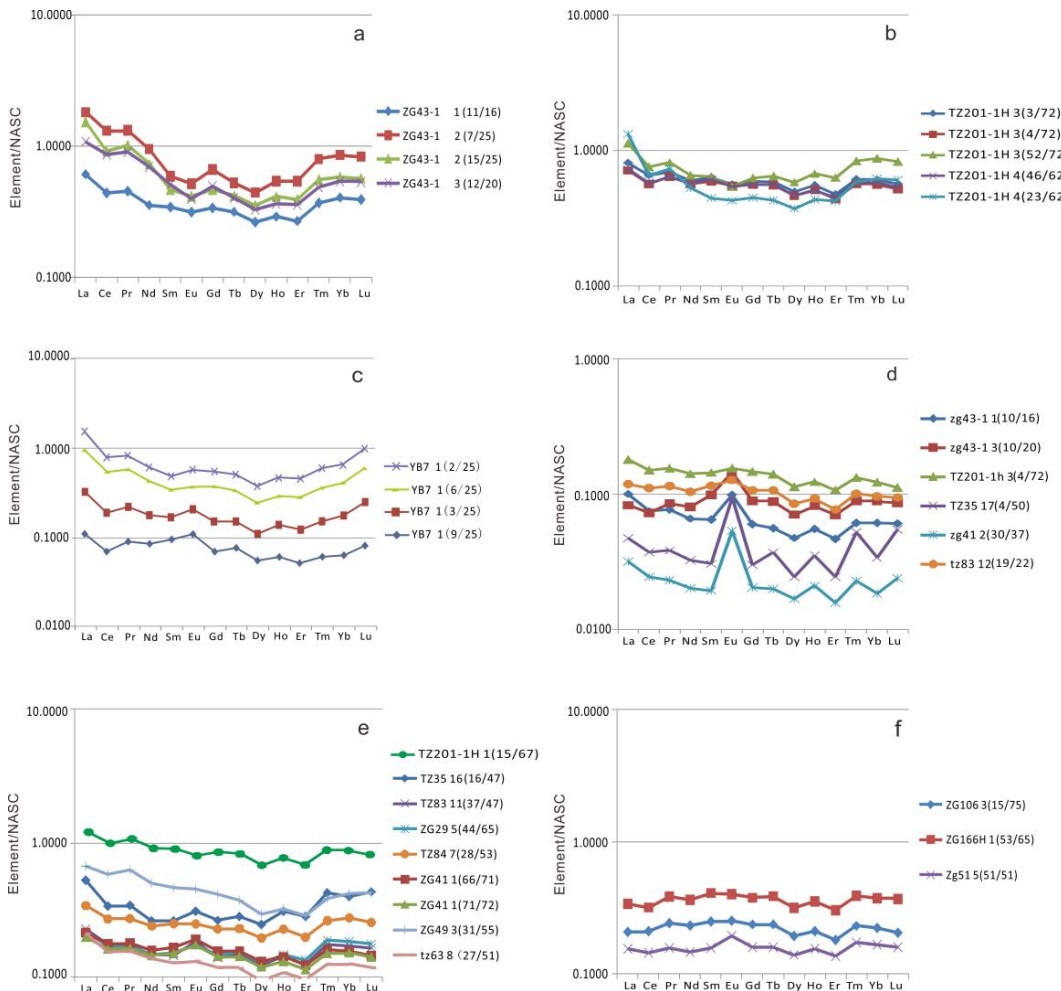

**Figure 10.** Rare earth distribution patterns of fillings in different layers or fracture-caves. (**a**) REE distribution pattern of the fracture-cave fillings in the LG43-1 Yingshan Formation is rightward, with negative Eu anomalies (δEu average 0.82); (**b**) REE distribution pattern of fracture-cavity fillings in TL201-1h Yingshan Formation, slightly right-tilt, slightly negative Eu anomaly (δEu average 0.91); (**c**) REE distribution pattern of the fracture-cave fillings in the YB7 Yingshan Formation is 'V' type, with slightly positive Eu anomalies (δEu average 1.05) (according to Wu et al. [27]); (**d**) REE distribution pattern of Yingshan Formation limestone, slightly right tilt, Eu positive anomaly (δEu average 1.32); (**e**) REE distribution pattern I of limestone at the bottom of Lianglitage Formation, micro-right-dip type, and positive Eu anomaly (δEu average 1.11); (**f**) REE distribution pattern II of limestone at the bottom of Lianglitage Formation, flat, and no Eu anomaly.

## 5. Discussion

### 5.1. Discussion on the Source of Mud Fillings in the Karst Cave

The mud fillings are developed in the fractures and caves in the northern slope of Tazhong and clarifying the sources of these fillings plays an important role in understanding the filling period and the filling environment and in searching for the unfilled oil and gas reservoirs. Before the study of this paper, Wu et al. [27], Sun et al. [25], and others analyzed the buried hill area of the Tarim Basin (the Carboniferous is over the Middle-Lower Ordovician, and the exposure time in this area is greater than 100 Ma) by means of petrology, palynology, and geochemistry [25,27]. It was found that the fillings of the Yingshan Formation in the cave are Devonian–Carboniferous sandstone, mudstone, and palynology, which belong to the Hercynian–Calidon long-term exposed fillings. However, for the northern slope area of central Tazhong (the Lianglitage Formation covered above the Yingshan Formation, and the exposure time in this area was less than 10 My), the Yingshan Formation has a shorter exposure time and less weathered and eroded strata, and the source of mud filling in the Yingshan Formation has been unclear.

According to the identification basis of Huang et al. on the clay composition and the cathodoluminescence of the paleo-weathering crust [49,50], the cathodoluminescence of limestone particles in the TL201-1H cave filling is bright, and the kaolinite in the fillings reflects the fact that the northern slope of Tazhong has experienced strong supergene dissolution and formed a rich dissolution space. In this study, we selected several cave fillings for pollen analysis, but unlike the previous Devonian–Carboniferous sporopollen found in the Yingshan Formation caves in the buried hill area [25,27], no pollen was found in the fillings of the northern slope of Tazhong. It indicates that there may have been very little terrigenous material mixed in the caves during the exposure period, or it proves that the exposure time was in the Early-Middle Ordovician, when terrestrial plants had not appeared on the land. Different from the karst fillings in the buried hill area studied by Wu et al. [27] and Sun et al. [25], further thin-section identification found that terrigenous quartz, feldspar, and other minerals were not found in the karst cave fillings of the two wells, but were mainly the mixture of carbonate debris or breccia, clay minerals, and calcite cements, indicating that there was no clastic rock or volcanic rock material source in the exposure period. The carbonate debris or breccia components were mainly micrite and lithic limestone by thin-section identification, and their lithology is consistent with the bedrock lithology of the Yingshan Formation. Zhang et al. [22] found that the carbon and oxygen isotope characteristics of the breccia are consistent with the bedrock of the Yingshan Formation [22]. It proves that the carbonate debris or breccia in these fillings come from the limestone of the Yingshan Formation and were formed by the dissolution of the debris or by collapse [22]. It is more likely that the result is of in situ dissolution. As for the source of calcite cements in the fillings, Dan et al. [20] and Zhang et al. [22] have found that their carbon and oxygen isotope values are basically consistent with limestone, indicating that they belong to syngenetic or quasi-syngenetic products [20,22]. In addition, the cathodoluminescence of the calcite in the fillings of the two wells in this paper shows that they do not emit light; so, they were likely to have settled from the early seawater of the Yingshan period rather than from meteoric water or spontaneous formation during diagenesis. As for the source of a large amount of clay, further analysis was needed.

The rare earth content in clastic rocks is mainly controlled by the rock composition of the provenance and reflects the geochemical characteristics of the provenance [51,52]. Therefore, rare earth elements can be used as an important source tracer. By comparing the REE distribution patterns of the fracture-cave fillings of the Yingshan Formation in two wells in the northern slope of Tazhong, the marlstone at the bottom of the Lianglitage Formation, and the limestone of the Yingshan Formation, as well as collecting the rare earth data of the fracture and cave fillings of the Yingshan Formation in the YB7 well in the buried hill area of the central Tarim Basin [27], the rare earth distribution pattern map (Figure 10c) can be made. Firstly, it was found that the rare earth distribution curves of the LG43-1 and TL201-1 H fracture-cave fillings were basically the same (Figure 10a,b), which shows that they have a unified material source. Secondly, the fracture-cave fillings in the two wells were different from those in the Yingshan Formation of Well YB7 in the buried hill area (Figure 10c). The distribution pattern of the Yingshan Formation in Well YB7 was a 'V' type (Figure 10c), showing a slight enrichment of the light rare earth, especially La, with an obvious negative Ce anomaly and a slight positive Eu anomaly (δEu average 1.05), indicating that the fillings of LG43-1 and TL201-1H were different from those of Well YB7. This also proves that Wu et al.'s inference that the karst cave fillings of YB7 are derived from Devonian sandstone [27]; so, they should belong to the fillings of a different exposure period. In addition, it is worth noting that the REE distribution curves of the fillings of the two wells are also quite different from those of the limestone of the Yingshan Formation (Figure 10a,b,e). According to the above test results, the ΣREE, the LREE/HREE, and the other indicators are also quite different (Figure 8). From the perspective of the argillaceous content (Figure 7), the limestone of the Yingshan Formation is relatively pure, and its argillaceous content is low; so, it is not likely that a large number of mud fillings in the cave originate from the Yingshan Formation. Finally, it was found that the REE distribution

pattern I of the fracture-cave fillings of the two wells is similar to that of the marlstone at the bottom of the Lianglitage Formation, and the similarity is high. The individual difference is mainly δEu1. Su et al. believed that Eu was a variable valence element [48]. Affected by the environment, the oxidation environment easily caused the Eu loss of the mud fillings in the cave during the supergene period.

The above characteristics show that the REE distribution curve of the karst cave fillings is similar to that of the bottom marlstone of the Lianglitage Formation, indicating that there is a large genetic relationship between them and reflecting that the fracture-cave fillings should be mainly derived from the marlstone of the Lianglitage Formation; it should be confirmed that the filling period is Caledonian, and that there was no terrestrial plant pollen in this period.

*5.2. Filling Paleoenvironment*

With negative δEu anomalies in the cave fillings and the strong cathodoluminescence of the limestone particles, the V/Cr, Ni/Cr, and V/V + Ni values of the cave fillings were less than 2, 5 and 0.75, respectively, indicating that the cave fillings were filled in exposed or shallow water oxidation environments. Dan et al. found that the cathodoluminescence of calcite symbiosis with mud in the cave in the northern slope of Tazhong did not show luminescence, and the carbon and oxygen isotope values showed that the calcite precipitated in the eogenetic marine environment [20].

Through the analysis of the Sr/Ba and B/Ga ratios of the cave fillings, we infer that the TL201-1H cave fillings were formed in seawater. The LG43-1 fillings were formed in brackish water or mixed water. The two geochemical indicators of the filling environment are inconsistent, which may be related to the clay content. However, the strong cathodoluminescence of the limestone particles in the TL201-1H fillings shows that the particles experienced strong meteoric freshwater dissolution, and the calcite in the two wells did not emit light, indicating that they were formed in seawater.

Therefore, considering the characteristics of the petrology, geochemistry and cathodoluminescence of the two wells, it is considered that the cave fillings of the Yingshan Formation in the northern slope of Tazhong can be divided into two stages:

(1) The exposed karst period of the Yingshan Formation: The Yingshan Formation was soon exposed by the tectonic movement of Caledonian Episode I, belonging to the large carbonate platform exposure. There was no clastic rock and volcanic material during the exposure period. Due to exposure to the Middle Ordovician before the appearance of terrestrial plants, no sporopollen was found. After the dissolution of the meteoric fresh water and the limestone dissolution and dissociation of the Yingshan Formation, clay and limestone particles were formed, which were filled into the karst caves by the meteoric fresh water (Figure 11a). The differences of dissolution and fillings between LG43-1 and TL201-1H are related to the location of the paleogeomorphology and the karst zone in the karst caves. According to the petrological differences of fillings and modern karst studies, it is speculated that the paleogeomorphology position of LG43-1 is lower than that of TL201-1H. In addition, the TL201-1H cave near the top of the Yingshan Formation belongs to the surface karst zone, and the LG43-1 cave is located in the karst runoff zone. The TL201-1H cave is more likely to form karst breccia in the surface karst zone, and the clay will seep downward, resulting in the low clay content of the cave, while the LG43-1 cave is located in the strong runoff zone, with more thorough dissolution and less breccia content. This is also the reason why the mineral contents in the fillings are quite different in different wells or in different cave intervals in the same well.

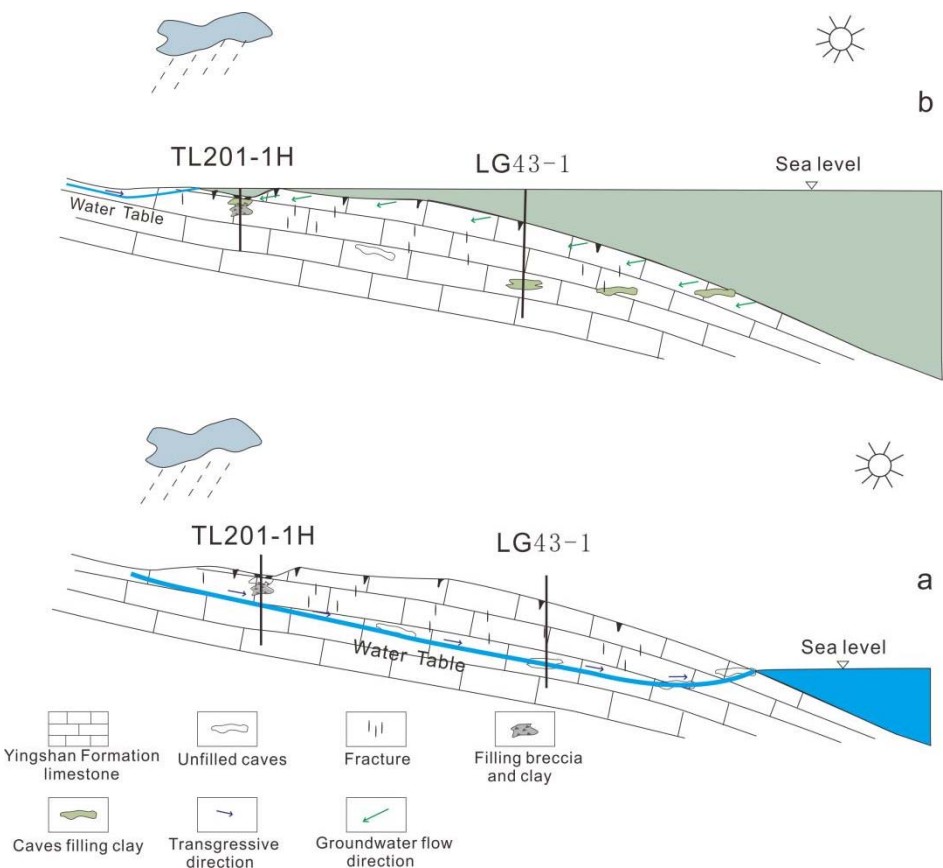

**Figure 11.** Karst cave filling pattern of Yingshan Formation in the northern slope of Tazhong. (**a**) The exposed period of Yingshan Formation is dominated by karstification, karst breccia, and clay filling into the cave; (**b**) during deposition of the Lianglitage Formation after exposure, seawater invaded the cave, and mud was brought into the filling cave with seawater.

(2) The transgression filling period after exposure: Sea water invaded during the deposition of the Lianglitage Formation, and the sea water further strongly affected the caves to form mixed water corrosion after exposure. Due to the rapid uplift of the northern Tarim Basin in the early deposition of the Lianglitage Formation, the volume of terrestrial material entering the ocean increased. The clay content was high, which also led to the filling of these clays by sea water into the caves (Figure 11b). The rare earth test showed that the cave fillings were mainly derived from the marlite at the bottom of the Lianglitage Formation. In addition, when seawater carrying argillaceous material entered the cave to fill it, calcite also settled with the clay due to supersaturation and settled along with the other materials. This shows that the filling effect of the Lianglitage Formation during transgression is the main source of the mud fillings in the Yingshan Formation karst cave on the northern slope of Tazhong, which has a wide range of influences and leads to the deterioration of the karst reservoir.

## 6. Conclusions

It was found by the sampling and analysis of the fracture-cave fillings in the Yingshan Formation that the fracture-cave fillings are mainly composed of dissociation particles in the bedrock of the Yingshan Formation, clay minerals, and calcite cements, and the fracture-cave fillings should mainly have come from the early seawater of the deposition of the Lianglitage Formation. There may be two filling effects. The first is that the limestone of the Yingshan Formation experienced the formation of karst caves due to meteoric freshwater dissolution during the exposure period, and the limestone of the Yingshan Formation was dissolved, resulting in some insoluble clay, residual limestone gravel particles, and cave-

collapse limestone gravel brought into the cave for filling. The second is the transgression of seawater during the deposition of the Lianglitage Formation. The seawater further strongly modified the caves. The clay content of the seawater in the early deposition of the Lianglitage Formation was high, which led to the further filling of these clays by seawater during the deposition of the Lianglitage Formation. It reflects the formation and karst cave fillings in the northern slope of Tazhong, which are related to freshwater or mixed water dissolution and seawater filling. On the whole, it is affected by the meteoric freshwater dissolution during the exposure period and the sea level fluctuation during the deposition of the Lianglitage Formation.

**Author Contributions:** Conceptualization, Y.D. and G.N.; methodology, Y.D.; formal analysis, Y.D.; investigation, J.L., S.J. and Q.Z.; writing—original draft preparation, Y.D.; writing—review and editing, G.N. and H.D.; project administration, B.L. All the co-authors performed a critical revision of the intellectual content of the paper. All authors have read and agreed to the published version of the manuscript.

**Funding:** This research was funded by the Geological Survey Program of the China Geological Survey (No. DD20190723), the National Key R&D Program of China (No. 2018YFC0604301), the Nonprofit Industry Research Program of the Chinese Academy of Geological Sciences (No. YYWF201723) and the Natural Science Foundation of Guangxi, China (No. 2018AD19040, No. 2018JJB150094, No. 2020GXNSFAA297095).

**Data Availability Statement:** Data is contained within the article.

**Acknowledgments:** Pan Wen-Qing and Zhang Zheng-Hong of the Research Institute of Exploration and Development at Tarim Oilfield Company of PetroChina have been of great assistance during the core observations and sample collection processes. Deng Zhen-ping (Manager) and Senior Engineer Yang Hui of the Karst Geology Resources and Environment Test Center of the Ministry of Land have been of great support during the sample testing and analysis processes.

**Conflicts of Interest:** The authors declare no conflict of interest.

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
