# Peer review of "The Source of Fracture-Cave Mud Fillings of the Ordovician Yingshan Formation and Its Paleokarst Environment in the Northern Slope of the Tazhong Uplift, Tarim Basin, China: Based on Petrology and Geochemical Analysis"

_minerals, doi:10.3390/min11121329_

Round 1
Reviewer 1 Report
This study deals with the origin of fracture-cave mud fillings and paleokarst environment of the Ordovician Yingshan Formation in the Northern Slope of the Tazhong Uplift, Tarim Basin, China. Based on petrology and geochemical analysis, the authors suggested that the fracture-cave fillings are mainly composed of the mixture of bedrock dissolution dissociation particles, clay minerals and calcite cements and the content of each component in different wells or cave interval is quite different. According to rare earth element composition, the fracture-cave fillings is similar to the bottom marlstones, indicating that the fracture-cave fillings should be mainly derived from the early sedimentary seawater sediments. Cathodoluminescence, trace element analysis and previous studies indicate that the formation and fillings of fractures and caves mainly occur in the oxidation environment of the hypergene period. The authors pointed out that the formation mechanism of karst cave reservoir in Yingshan Formation has important theoretical significance for guiding the next oil and gas exploration in this area. This study is interesting and may interest to the broad readers. The geological, petrographical and geochemical data were given in detail, but the mineralogical parts is insufficient and some points needs to clarified and some additional data is necessary as noted in the attached manuscript file. After revision of these points, the manuscript could be considered by editor for acceptance.
Some comments/suggestions were given below:
- The mineralogical composition of the studied mud-filling materials should be given as bulk composition (XRD data)? The origin of clay minerals should be given definitely (detrital particles?, inherited from host-rock? Authigenic (or cement) occurrences?)
- Some sentences are unclear and they should be clarified, i.e., “sedimentary sea-water”, “oxidation environment of the hypergene period”
- XRD method was used for only clay minerals. The representative whole-rock and clay fraction mineralogical data (XRD patterns) of two wells should be given for comparison. What means interstratified ratio (%S)? You should define interstratification type (R0, R1, R2 etc..) before the smectite contents of I/S and C/S. These data looks not necessary, because they did not use for this study.
- What is the reason of positive or negative Eu anomaly? Which minerals or environmental conditions etc. are responsible for this?
Author Response
Dear Editor,
Thank you very much for your letter and the comments from the reviewers about our paper submitted to The source of fracture-cave mud fillings of the Ordovician Yingshan Formation and its paleokarst environment in the Northern Slope of the Tazhong Uplift, Tarim Basin, China: Based on petrology and geochemical analysis (Manuscript ID: minerals-1416606).
We have checked the manuscript and revised it according to the comments. We submit here the revised manuscript as well as a list of changes.If you have any question about this paper, please don’t hesitate to let me know.
Sincerely yours,
Dr.Nie
Response to Reviewer 1:
Thanks for your comments on our paper. We have revised our paper according to your comments:
1.The mineralogical composition of the studied mud-filling materials should be given as bulk composition (XRD data)? The origin of clay minerals should be given definitely (detrital particles?, inherited from host-rock? Authigenic (or cement) occurrences?)
Answer: The mineral composition of mud-filling materials were mainly estimated through thin slice identification, so X-ray diffraction is not performed. In addition, the chemical composition of the mud-filling materials were estimated in this paper. The research purpose can be achieved by the above two methods.
In addition, according to your comments, the source of clay minerals was clarified. Clay materials mainly came from the ocean where a large amount of marlstone limestone were deposited in the early Lianglitage. Due to the uplift of the northern Tarim structure in the early Lianglitage, a large amount of terrigenous clastic materials were injected into the ocean. These clay materials are brought into sea caves by transgression. Supplementary explanation has been added in the discussion section and marked out in the revised paper.
- Some sentences are unclear and they should be clarified, i.e., “sedimentary sea-water”, “oxidation environment of the hypergene period”
Answer: It has been modified that the formation and fillings of fractures and caves mainly occur in the hypergene period that has the characteristic of oxidized environment.
3.XRD method was used for only clay minerals. The representative whole-rock and clay fraction mineralogical data (XRD patterns) of two wells should be given for comparison. What means interstratified ratio (%S)? You should define interstratification type (R0, R1, R2 etc..) before the smectite contents of I/S and C/S. These data looks not necessary, because they did not use for this study.
Answer:In this paper, the different types of clay mineral combinations in Well TZ201-1H and LG43-1 have been mainly used to determine that the fillings had experienced diagenetic environment. The discovery of a small amount of kaolinite in TH201-1h indicates that it has experienced an acidic karst environment on the surface and represents paleokarstification. The interstratified ratio (%S) in Table 1 is useless in this article, it is deleted.
4.What is the reason of positive or negative Eu anomaly? Which minerals or environmental conditions etc. are responsible for this?
Answer:The fillings of the fractures and caves in the two wells are δEu<1, while the marlstone at the bottom of the Lianglitage Formation is δEu>1. According to Su et al. (2012) research believes that Eu is a variable element, affected by the environment, Eu2+ in the oxidizing environment in the epigenetic stage is easily oxidized to Eu3+, which leads to the loss of Eu in the fillings. On the other hand, for minerals, calcium-rich minerals affect the differentiation of Eu2+. Calcium-rich minerals are prone to isomorphism with Eu2+ and generally show a positive anomaly. Therefore, the Yingshan Formation limestone with more calcium and the marls of the Lianglitage Formation (Cao about 55%) all show positive anomalies. It is worth noting that although the CaO of mud fillings (Cao about 35~45%) is relatively less than that of the limestone, the CaO content is actually higher, and the influence of calcareous minerals is only considered, so that mud fillings have not a negative anomaly. Therefore, we believe that the negative anomaly of the cave fillings are that the environmental conditions contribute more and represent the oxidizing environment. Supplementary explanation has been added in the discussion section and marked out in the revised paper.
Reviewer 2 Report
Please see my comments file.

Author Response
Dear Editor,
Thank you very much for your letter and the comments from the reviewers about our paper submitted to The source of fracture-cave mud fillings of the Ordovician Yingshan Formation and its paleokarst environment in the Northern Slope of the Tazhong Uplift, Tarim Basin, China: Based on petrology and geochemical analysis (Manuscript ID: minerals-1416606).
We have checked the manuscript and revised it according to the comments. We submit here the revised manuscript as well as a list of changes.If you have any question about this paper, please don’t hesitate to let me know.
Sincerely yours,
Dr.Nie
Response to Reviewer 2:
Thanks for your comments on our paper. We have revised our paper according to your comments:
1.In the annotated file are shown both corrected errors as well as the sentences that need rephrasing, mostly because their logic suffers from the jargon-related mental shortcuts.
Answer: According to your comments, errors as well as the sentences that need rephrasing in the annotated file have been checked and revised,and marked out in the revised paper.
2.The introductory chapters (Introduction and Geological setting) are too long - please shorten them by a half so that your message becomes more clear.
Answer: According to your comments, the introduction and geological background has been shortened and rewritten, and marked out in the revised paper.
Reviewer 3 Report
In the annotated file are shown both corrected errors as well as the sentences that need rephrasing, mostly because their logic suffers from the jargon-related mental shortcuts.
The introductory chapters (Introduction and Geological setting) are too long - please shorten them by a half so that your message becomes more clear.

Author Response
Dear Editor,
Thank you very much for your letter and the comments from the reviewers about our paper submitted to The source of fracture-cave mud fillings of the Ordovician Yingshan Formation and its paleokarst environment in the Northern Slope of the Tazhong Uplift, Tarim Basin, China: Based on petrology and geochemical analysis (Manuscript ID: minerals-1416606).
We have checked the manuscript and revised it according to the comments. We submit here the revised manuscript as well as a list of changes.If you have any question about this paper, please don’t hesitate to let me know.
Sincerely yours,
Dr.Nie
Response to Reviewer 3:
Thanks for your comments on our paper. We have revised our paper according to your comments:
1.Some parts of this manuscript do not have enough scientific detail from a mineralogical or sedimentological perspective, especially for a journal of Minerals. For example, there is not a good description of the sedimentary depositional setting for the Yingshan or Lianlitage formations. Were they carbonate ramps, platforms, etc.?
Answer: According to your comments, The sedimentary background description has been added in the chapter of geological background, and marked out in the revised paper.
2.Also, Line 227, for example reads “gravels have certain sorting”. What does this mean? What is the sorting?
Answer:It has been modified to poor sorting
3.Other terms like “soft and dirty” or “almost no dirty” are nonsensical and non-scientific. Are they describing organic carbon content? The reader should not be forced to guess the meanings.
Answer:It has been modified to result in different degrees of looseness and compactness of the fillings due to the difference in calcium content. Delete "soft and dirty" description
4.None of the chemical formulae in the manuscript body or figures have subscripts in the appropriate locations. This is not acceptable.
Answer: Checked and revised, and marked out in the revised paper.
5.Trace element arguments in Lines 292-314. Lines 292-293 describe “studies”, but there are no citations. Only one 42-year old citation is given in the following paragraph. This section seems like a stretch, and it is not convincing that this is the only mechanism to cause variations in these elemental ratios.
Answer: New references have been added, and marked out in the revised paper.
Response to Reviewer 3 Line-by-Line revised:
1.Line 16: extra spaces
Answer:Checked and revised, and marked out in the revised paper.
2.Line 32: into the caves
Answer:Checked and revised, and marked out in the revised paper.
3.Line 39-40: I don’t understand what “into the buried underground” means. Does it mean “into the subsurface”?
Answer:Yes, they were revised and marked out in the revised paper.
4.Line 44-45: This sentence needs clarification. There should be author names as nouns where the [14,15] is present.Line 51 and elsewhere: The use of “serious” here is not appropriate. I suggest “widespread” or “extensive”.
Answer:According to your comments, they were revised and marked out in the revised paper.
5.Lines 56-73: This is too long and wordy and needs to be condensed. In particular,Lines 69-71 are unclear. Line 69: “oxygen isotope methods”
Answer:According to your comments, they were condensed ,revised and marked out in the revised paper.
6.Lines 75-82: Shorten this and either reword or delete lines 77-79.
Answer:According to your comments, delete lines 77-79.
7.Line 90: I suggest “area was the middle” ,
Answer:Checked and revised,and marked out in the revised paper.
8.Line 94: I suggest “exploration has been concentrated”
Answer:Checked and revised,and marked out in the revised paper.
9.Line 97. Episode?
Answer:Checked and revised,and marked out in the revised paper.
10.Line 100: “Seriously” should be changed to “deeply”?
Answer:Checked and revised,and marked out in the revised paper.
11.Line 104: “The age of the Yingshan”.
Answer:Checked and revised,and marked out in the revised paper.
12.Line 107: delete “a set of”.
Answer:Checked and revised,and marked out in the revised paper.
13.Lines 109-110: These sentences should be combined so Yingshan Formation is not repeated. This type of repetitive writing occurs throughout the manuscript.
Answer:According to your comments, These sentences has been combined and marked out in the revised paper.
14.Figure 1. This figure is good overall, but the font size is WAY too small. I can’t read it when it is printed, even with glasses. Even when I enlarge on my large monitor, some of the words overlap the lines in the figure. Furthermore, the stratigraphy here is for China only, yet this is an international journal. I recommend that part (b) of this figure be expanded into its own figure so it is readable, along with international stratigraphic nomenclature. The International Commission on Stratigraphy is a good place to start: https://stratigraphy.org/chart
Answer:According to your comments, part (b) of the figure 1 has been expanded into new figure 2 and marked out in the revised paper
15.Line 120 and Section 2.2 on this page: A new section cannot start with “However,”. That’s only for a segue between thoughts in the same section. The entire paragraph from Line 120 to 144 needs to be rewritten. It is confusing overall, and it includes terms that I don’t understand.
Answer:Checked and rewritten,and marked out in the revised paper.
16.What is “dark block”? Define terms like GR as Gamma-Ray at least once for the reader. Only 2 coring wells with high GR intervals show mud fillings. Do the other coring intervals not have mud fillings in the high GR intervals? Or are there only 2 cored intervals? Or something else? It’s just not clear. Also, “58.07% of the filling interval is below the unconformity surface 40 m” is not clear. Do the authors mean that 58% of the available microporosity within 40 m below the unconformity was filled by these mud fillings? Or do they mean that 42% of the fillings are
above the unconformity? It’s just not clear.
Answer:Section 2.2 has been checked and rewritten, and marked out in the revised paper.
17.Line 140 has “large width and higher ratio karst caves” is unclear. I have no idea what this means.
Answer: The sentences has been deleted and marked out in the revised paper.
18.Line 143: Is it really “urgently” necessary? Why? Where the caves are filled is clearly important – I understand that, but tell the reader why the source of the clays is so important to know. How does that help with petroleum exploration? Wouldn’t just an empirical mapping of these systems be more valuable?
Answer:Checked and rewritten, and marked out in the revised paper.
19.Figure 2: This is not the pattern of Swiss cheese (although people do commonly say that). Swiss cheese has isolated pores that are not connected to each other due to bacterial decomposition. Karst systems by their nature have interconnected pores and are therefore not like Swiss cheese.
Answer:According to your comments, These sentences about pattern of Swiss cheese have been deleted and marked out in the revised paper.
20.Figure 3. This figure is poorly constructed and labeled. It does not look like the quality necessary for an international journal. It’s blurry, the “Depth below unconformity” label should be vertically oriented near the Y-axis, and there is a floating 30 with a strike and dip symbol near the upper left. Font sizes also vary. The caption is also unclear. There is a (b) label, but no (a) label in the figure or the caption.
Answer:This figure has been revised to new figure 4 and marked out in the revised paper.
21.Section 3.4 heading is the same as 3.3. Is this supposed to be for REE?
Answer:Checked and revised,and marked out in the revised paper.
22.Line 199. The words “precipitated” and “distilled” do not seem correct here. Do the authors mean “settled” and “decanted”? It’s confusing.
Answer:Yes, it has been checked and revised,and marked out in the revised paper.
23.Line 204: In English, a thermostat is a small device to control temperature. You cannot put anything inside of one. Do the authors mean “oven”?
Answer:Yes, it has been checked and revised,and marked out in the revised paper.
24.Section 4 (Results): I will just summarize here. The descriptions here are not precise enough for a peer-reviewed journal. I don’t mean that they’re too short. I mean that the terms are not precise. Which minerals are present (calcite, aragonite, dolomite)? These minerals were potentially in the host rock formations, yet I see no mention beyond calcite or “limestone”.
Answer:The means that the fracture-cave mud fillings is mainly composed of limestone dissolution collapsed gravel or dissolution residual material, clay minerals and calcite cements in section4.1.
25.Also, Line 227, for example reads “gravels have certain sorting”. What does this mean? What is the sorting? Other terms like “soft and dirty” or “almost no dirty” are nonsensical and nonscientific. The authors use 3 different versions of the word for CL, including cathodoluminescence, cathode luminescence, and cathodic luminescence. Pick one.
Answer: “gravels have certain sorting” has been rewritten “gravels have poor sorting”.These terms like “soft and dirty” or “almost no dirty” have been deleted. According to your comments, Pick ‘cathodoluminescence’for CL and marked out in the revised paper.
26.Lines 245-249: It seems like this is a conclusion. Doesn’t it belong in the Discussion section at least? It’s better to state that the data are “consistent with” the particles originating from the Yingshan Formation, rather than stating it as a fact.
Answer:According to your comments, the paragraph has been deleted and put it in the discussion section
27.Line 286: Caves.
Answer:Checked and revised,and marked out in the revised paper.
28.Lines 292-314. I am simply not convinced by this section. Lines 292-293 describe “studies”, but there are no citations. Only one 42 year old citation is given in the following paragraph. This section seems like a stretch, and it is not convincing that this is the only mechanism to cause variations in these elemental ratios.
Answer:New references have been added,and marked out in the revised paper.
29.Line 416: strata and the source
Answer:Checked and revised,and marked out in the revised paper.
30.Line 448: is this the only possibility? What if there were different sources for these 2 locations, such as different rivers, etc.
Answer:That is impossible, because Wang (2017) and other studies believe that the filling of YB7 is Devonian and belongs to the Hercynian exposed filling. The fillings of the two wells in this paper are from the Late Ordovician period and should belong to the Caledonian period. The northern slope area studied in this paper is not exposed during the Hercynian period. So they are exposed to different periods, and they cannot be different rivers in the same period.
31.Line 514. This is incorrect and impossible. A solid object cannot be a dissolved object. If it is dissolved it would be in solution. Do the authors mean insoluble residues?
Answer:Limestone dissolution collapsed gravel or dissolution residual material
32.Line 521: Instead of atmospheric, I think “meteoric” is a more appropriate term here.
Answer:Checked and revised,and marked out in the revised paper.
Round 2
Reviewer 1 Report
The revision made on manuscript, but XRD bulk analysis ( determinations) could be better than thin-section interpretations especially for fine- grained components.
Author Response
Dear Editor,
Thank you very much for your letter and the comments from the reviewers about our paper submitted to The source of fracture-cave mud fillings of the Ordovician Yingshan Formation and its paleokarst environment in the Northern Slope of the Tazhong Uplift, Tarim Basin, China: Based on petrology and geochemical analysis (Manuscript ID: minerals-1416606).
We have checked the manuscript and revised it according to the comments. We submit here the revised manuscript as well as a list of changes. If you have any question about this paper, please don’t hesitate to let me know.
Sincerely yours,
Dr. Nie
Response to Reviewer 1:
Thanks for your comments on our paper. We have revised our paper according to your comments:
- The revision made on manuscript, but XRD bulk analysis (determinations) could be better than thin-section interpretations especially for fine- grained components.
Answer: You are right, XRD bulk analysis (determinations) is better than thin-section interpretations especially for fine- grained components. But I do not have more samples for testing and analysis, coupled with the epidemic and financial constraints, can not go to Xinjiang again for sampling, so only use thin-section interpretation.
Reviewer 2 Report
Please see my review. The manuscript is improved, but there are a few items that could still be improved.

Author Response
Dear Editor,
Thank you very much for your letter and the comments from the reviewers about our paper submitted to The source of fracture-cave mud fillings of the Ordovician Yingshan Formation and its paleokarst environment in the Northern Slope of the Tazhong Uplift, Tarim Basin, China: Based on petrology and geochemical analysis (Manuscript ID: minerals-1416606).
We have checked the manuscript and revised it according to the comments. We submit here the revised manuscript as well as a list of changes. If you have any question about this paper, please don’t hesitate to let me know.
Sincerely yours,
Dr. Nie
Response to Reviewer 2:
Thanks for your comments on our paper. We have revised our paper according to your comments:
- In several places, there is a new term used in the manuscript “Lianglitage period” that seems problematic. Although “period” is lower case, it is still a bit confusing because there is no formal geologic period, and “Lianglitage” is a formally named formation. The period in geology would be strictly defined as “Ordovician”. The Katian stage could be used for a time, but that may not be preferred. What about phrasing it something like “during deposition of the Lianglitage Formation”?
Answer: Checked and revised, and marked out in the revised paper.
- There is one new error introduced in line 117. The age of these two formations is not 12-15 My. I think the authors are trying to describe the duration of deposition. The age is Ordovician (455 million years ago). Plus there are other units between them. Perhaps something like “The length of time represented by deposition from the start of the Yingshan Formation through the end of the Lianglitage Formation is about 12-15 million years.”
Answer: I try to describe that the exposure duration of Yingshan Formation is about 7 – 8 My
- It might be useful to describe who did the pollen analysis. There are palynomorphs present during this part of Earth’s history, but I don’t think true pollen was around yet. It’s not clear what the lack of pollen means, but a sentence on this is important. Is it missing because of the age of the material, or perhaps this was marine mud or an influx of mud from an area without land plants. This is not a major issue, but as the reader, I was curious.
Answer: We believe that the main reason is that the exposure time was in the Middle Ordovician, when the terrestrial plants were underdeveloped on the Earth. Even if it is later filled by marine mud, there will always be a small amount of terrestrial mud mixed into the cave during the exposure period of the Yingshan Formation, and sporopollen may be found. Therefore, the marine mud in the karst cave may not be the main reason for not finding the sporopollen; In addition, previous studies [25][27] found that the Devonian-Carboniferous sporopollen in the karst cave fillings of the Yingshan Formation in the buried hill area, indicating that the buried area was exposed in the Devonian-Carboniferous. We know that there were not a large number terrestrial plants of Middle Ordovician until in Devonian in the Earth’s history. Therefore, there is no sporopollen in the karst caves in the north slope area of our paper, which is most likely caused by the exposure time in the Middle Ordovician, which is also consistent with the Ordovician strata above and below the exposed surface.
Line 31: Run-on sentence. Perhaps “filling. Second, seawater” might fix this
Answer: Checked and revised, and marked out in the revised paper.
Lines 47-49: The grammar needs to be fixed here
Answer: Checked and revised, and marked out in the revised paper.
Line 60: In a new paragraph, “it” is unclear.
Answer: Checked and revised, and marked out in the revised paper.
Line 63 Line 66: I suggest keeping the year or the reference number when these authors are listed.
Line 76: “a mixt water environment”
Answer: Checked and revised, and marked out in the revised paper.
Line 109: In this case, I think “late” should be capitalized before Ordovician. The Lianglitage is
still not a period.
Answer: Checked and revised, and marked out in the revised paper.
Line 143: This is ok, but only a brief explanation is needed, such as “GR (Gamma-ray log) intervals”.
Answer: Checked and revised, and marked out in the revised paper.
Line 184: Still a bit unclear. How about “Among the 61 mud filling intervals identified in this
area, only the 3 mud filling 183 intervals of TZ201-1H and ZG43–1 occur in cored intervals.”
Answer: Checked and revised, and marked out in the revised paper.
Line 228: “X-Ray Diffraction analyses for clay mineral composition”
Answer: Checked and revised, and marked out in the revised paper.
Line 255: Calcite is “filled with” is unclear. Do the authors mean that the calcite contains
abundant solid inclusions of this material?
Answer: I means that calcite settles between these substances. Checked and revised, and marked out in the revised paper.
Lines 260-261: I still do not understand what the “grinding circle” is. I have never seen this
phrase before, and readers will be confused if it is not defined or reworded.
Answer: I means degree of roundness. Checked and revised, and marked out in the revised paper.
Line 390, etc. Should “Earth” be capitalized?
Answer: Checked and revised, and marked out in the revised paper.
Line 467: “dissolution” instead of “dis-solution”.
Answer: Checked and revised, and marked out in the revised paper.
Line 518: I suggest adding a phrase for clarity so it reads “cave fillings, we infer that”
Answer: Checked and revised, and marked out in the revised paper.
Line 551: This is unclear. Settling does not occur due to supersaturation. However, precipitation can occur, which could be followed by settling of the crystals. I suggest something like “calcite precipitated due to supersaturation and settled along with the other materials”. Why would the water entering the cave be supersaturated though? Due to curved saturation lines, mixing of any two saturated waters would produce an undersaturated water.
Answer: Checked and revised, and marked out in the revised paper
Reviewer 3 Report
I attach my remarks in the attached file of your paper.

Author Response
Dear Editor,
Thank you very much for your letter and the comments from the reviewers about our paper submitted to The source of fracture-cave mud fillings of the Ordovician Yingshan Formation and its paleokarst environment in the Northern Slope of the Tazhong Uplift, Tarim Basin, China: Based on petrology and geochemical analysis (Manuscript ID: minerals-1416606).
We have checked the manuscript and revised it according to the comments. We submit here the revised manuscript as well as a list of changes. If you have any question about this paper, please don’t hesitate to let me know.
Sincerely yours,
Dr. Nie
Response to Reviewer 3:
Thanks for your comments on our paper. We have revised our paper according to your comments:
- I attach my remarks in the attached file of your paper.
Answer: Checked and revised, and marked out in the revised paper.